

# Real-time evolution in the Hubbard model with infinite repulsion

Elena Tartaglia[1], Pasquale Calabrese[2,3] and Bruno Bertini[4⋆]

**1** Data61, CSIRO, 34 Village St, Docklands VIC 3008, Australia
**2** SISSA and INFN, via Bonomea 265, 34136, Trieste, Italy
**3** International Centre for Theoretical Physics (ICTP), I-34151, Trieste, Italy
**4** Rudolf Peierls Centre for Theoretical Physics, Oxford University,
Parks Road, Oxford OX1 3PU, United Kingdom

⋆ bruno.bertini@physics.ox.ac.uk

## Abstract

We consider the real-time evolution of the Hubbard model in the limit of infinite coupling. In this limit the Hamiltonian of the system is mapped into a number-conserving quadratic form of spinless fermions, i.e. the tight binding model. The relevant local observables, however, do not transform well under this mapping and take very complicated expressions in terms of the spinless fermions. Here we show that for two classes of interesting observables the quench dynamics from product states in the occupation basis can be determined exactly in terms of correlations in the tight-binding model. In particular, we show that the time evolution of any function of the total density of particles is mapped directly into that of the same function of the density of spinless fermions in the tight-binding model. Moreover, we express the two-point functions of the spin-full fermions at any time after the quench in terms of correlations of the tight binding model. This sum is generically very complicated but we show that it leads to simple explicit expressions for the time evolution of the densities of the two separate species and the correlations between a point at the boundary and one in the bulk when evolving from the so called generalised nested Néel states.



# 1 Introduction

The Hubbard model is the fundamental paradigm of strongly correlated electrons and, as such, is attracting the attention of theoreticians from many different corners of condensed matter. In particular, the 1D version of the Hubbard model is at the focus of the theoretical activity ever since Lieb and Wu discovered its Bethe Ansatz integrability [1]. In the context of many-body systems, however, integrability does not mean simplicity: the models solved by Bethe-Ansatz generically retain non-trivial interactions and many interesting quantities — most importantly correlations functions — remain inaccessible in most cases. Moreover, the Bethe Ansatz solution of the 1D Hubbard model is particularly complicated from the technical point of view: it involves two intertwined sets of Bethe equations, i.e. it is *nested* in the technical parlour, and its R-matrix is not in difference form [2]. In spite of this mathematical complexity, however, a remarkable collective effort led to a comprehensive description of its equilibrium thermodynamics culminating in a complete account of its rich phase diagram [2].

A natural next step is then to understand the 1D Hubbard model out of equilibrium. Indeed, a substantial body of work distributed over the last 15 years (see, e.g., the reviews in the volumes [3,4]) has revealed that, when driven out of equilibrium, integrable models display rich and interesting physics. For instance, in homogeneous settings, local observables in out-of-equilibrium integrable models relax to values described by the *generalised* Gibbs Ensemble (GGE) [5,6] rather than the standard thermal ensemble expected for non-integrable systems [7,8]. The GGE is a statistical ensemble built with all the local and quasi-local conserved charges of the Hamiltonian [9,10], which are extensively many in integrable models. Therefore, the GGE retains extensive memory about the initial configuration. The exact knowledge of the stationary state allows one to compute stationary values of observables without solving the overwhelmingly complicated many-body dynamics and, moreover, it also gives access the asymptotic dynamics of correlations [11] and entanglement [12–14]. Importantly, the occurrence of GGEs has also been observed experimentally [15].

Another remarkable breakthrough has been the discovery that the late time behaviour of integrable systems in inhomogeneous settings is described by a set of hydrodynamic equations involving all the quasi-local charges: such an unconventional hydrodynamic theory has been termed *generalised* hydrodynamics (GHD) [16–20]. Despite being macroscopically many, the GHD equations can be treated analytically, allowing for an unprecedented quantitative comparison between theory and experiments [21–23].

Very few of the aforementioned results, however, have so far been explicitly tested in the case of the Hubbard model. This is mainly because none of the analytical approaches developed to determine the stationary state of out-of-equilibrium integrable systems [24–27] can be applied to the Hubbard model with finite interaction strength due to its technical complexity. In essence, the evolution of the Hubbard model has been accessed only in the quasi-stationary regime using GHD equations [28, 29], whose solution has been explicitly worked out also for models with a nested Bethe ansatz [30–35]. A promising direction to compute stationary values in the Hubbard model is the so called *quench action* method [26, 27]. Indeed, this approach does not require the explicit form of the set of conserved charges and, for this reason, it has been used to obtain exact results even when, like in the Hubbard model, the full charge spectrum was unknown [36–49]. A limitation, however, is that this method requires the exact overlaps between the initial state and all the eigenstates of the time-evolving Hamiltonian. These quantities have been accessed only for a few special initial states [50–60], including one in the Hubbard model with finite interaction strength [60].

A more general open question concerns the finite-time dynamics of the Hubbard model. Namely the time evolution of the system away from the asymptotic regime. This regime is clearly very interesting — for instance accessing it could reveal the quantitative mechanisms allowing quantum many-body systems to attain local equilibrium — but is largely uncharted in integrable models due to the lack of methods — both analytical and numerical — which are able to access it. For instance, although in principle the quench action can provide the full post-quench dynamics [26], this task has to be performed numerically [61, 62] and can only access short times or small systems. In summary, because of its intrinsic difficulty, the non-equilibrium dynamics of the one-dimensional Hubbard model have so far been investigated mainly by approximate means, see e.g. Refs. [72–81].

Interestingly, however, in recent years a number of exact results concerning the finite time dynamics have been found in a special class of integrable systems that can be thought of as strong coupling limits of standard integrable models [63–71]. In this spirit, in Ref. [82] we considered Hubbard in the limit of infinite repulsion, when the thermodynamics becomes essentially that of a free model. Exploiting the quench action approach we built the stationary state reached after quenches from a family of relatively simple low entangled initial states. The infinite repulsion limit of the Hubbard model has later been considered in Ref. [83], which presented a method able to reconstruct the full time evolution of two-point correlations from a special class of product initial states where the spin has to flip from one particle to the next. The approach of Ref. [83] is based on a mapping to non-interacting spinless fermions, developed in Ref. [84], that works at the level of the correlations and accounts for their evolution numerically with polynomial complexity in time. In this paper we focus again on the finite-time dynamics of the Hubbard model in the limit of infinite repulsion but reconstruct the dynamics of relevant observables using a *different* mapping to free fermions. In particular we consider the operatorial mapping introduced by Kumar in Ref. [85, 86]. Applying Kumar's mapping we find explicit analytical expressions for certain relevant correlations. Differently from the results of Ref. [83], our findings can be applied for initial states with generic spin configurations and the explicit expressions that we provide can be evaluated in the thermodynamic limit. On the other hand, being valid for more general initial states, our expressions for *generic* two-point correlations are more complicated than those found in Ref. [83].

The rest of this manuscript is laid out as follows. In Sec. 2 we introduce the model and discuss the infinite interaction limit. In Sec. 3 we provide a detailed review of Kumar's mapping for the local observables in the infinite repulsion limit. Sec. 4 is the core of this manuscript where we use this mapping to obtain the time evolution of several local observables. In Sec. 5 we summarise our results and discuss some further developments. Two appendices contain some technical details of our calculations.

## 2 Hubbard Model with Infinite Repulsion

We consider a system of interacting spin-full fermions on a one-dimensional lattice whose dynamics are described by the Hubbard Hamiltonian [2], i.e.

$$H = -J \sum_{x=1}^{L-1} \sum_{\alpha=\uparrow,\downarrow} \left( c_{x,\alpha}^\dagger c_{x+1,\alpha} + c_{x+1,\alpha}^\dagger c_{x,\alpha} \right) + U \sum_{x=1}^{L} \left( n_{x,\uparrow} - \frac{1}{2} \right)\left( n_{x,\downarrow} - \frac{1}{2} \right). \tag{1}$$

Here we denoted by $\{c_{x,\alpha}\}$ a set of spin-1/2 fermionic operators fulfilling the canonical anti-commutation relations

$$\{c_{x,\alpha}, c_{y,\beta}\} = \{c_{x,\alpha}^\dagger, c_{y,\beta}^\dagger\} = 0, \qquad \{c_{x,\alpha}, c_{y,\beta}^\dagger\} = \delta_{x,y}\delta_{\alpha,\beta}, \qquad \alpha,\beta = \uparrow,\downarrow, \tag{2}$$

and by $n_{x,\uparrow/\downarrow}$ the local number operators

$$n_{x,\alpha} = c_{x,\alpha}^\dagger c_{x,\alpha}. \tag{3}$$

Moreover we indicated by $L$ the number of sites of the chain and we adopted open boundary conditions.

The vacuum state is defined in the usual fashion as the state annihilated by the operators $c_{x,\alpha}$

$$c_{x,\alpha}|0\rangle = 0, \qquad x = 1, 2, \dots, L, \qquad \alpha = \uparrow, \downarrow. \tag{4}$$

Any site on the lattice can have no particles, a fermion with spin up, a fermion with spin down or two fermions with different spin. These states at site $x$ are constructed by acting with the fermionic creation operators on the vacuum as follows

$$|0\rangle, \qquad c_{x,\uparrow}^\dagger|0\rangle, \qquad c_{x,\downarrow}^\dagger|0\rangle, \qquad c_{x,\uparrow}^\dagger c_{x,\downarrow}^\dagger|0\rangle. \tag{5}$$

Since at every site there are 4 possible states, the total Hilbert space has $4^L$ states. We refer to the states with at least one site with two particles as *double occupancy* states; note that these states are the only ones affected by the interaction term.

In this paper we are interested in the limit of infinite interaction, i.e., $U \to \infty$. This limit can be thought of as a limit of infinite repulsion among the fermions as no two particles can sit on the same site. Indeed, the double occupancy states have infinite energy and, therefore, become unphysical. The physical Hilbert space is then the $3^L$-dimensional subspace with no double occupancy states.

The effective Hamiltonian describing the dynamics in this limit is written as

$$H_\infty = -J\,\mathcal{P}\left[\sum_{x=1}^{L-1}\sum_{\alpha=\uparrow,\downarrow}\left(c_{x,\alpha}^\dagger c_{x+1,\alpha} + c_{x+1,\alpha}^\dagger c_{x,\alpha}\right)\right]\mathcal{P}, \qquad \mathcal{P} = \prod_{x=1}^{L}(1 - n_{x,\uparrow}n_{x,\downarrow}). \tag{6}$$

This Hamiltonian is known as the $t$-$0$ model [2] and is non-trivial only when the number of particles is less than $L$. We remark that, as shown in [87], the Hamiltonian (6) provides a fermionic representation of the so called Maassarani-Mathieu model [88] — a SU(3) general-isation of the XX spin chain.

## 3 Kumar mapping to free fermions

Interestingly, as shown by Kumar in Ref. [86] (see also Refs. [85,89]), the Hamiltonian (6) is exactly mapped into the tight-binding model by a unitary transformation, i.e.

$$\mathcal{U}^\dagger H_\infty \mathcal{U} = H_{\text{tb}} \equiv -J\sum_{x=1}^{L+1}(f_{x+1}^\dagger f_x + f_x^\dagger f_{x+1}), \qquad \mathcal{U}^\dagger\mathcal{U} = \mathcal{U}^\dagger\mathcal{U} = I, \tag{7}$$

where $\{f_x\}$ are canonical spinless fermions fulfilling

$$\{f_x, f_y\} = \{f_x^\dagger, f_y^\dagger\} = 0, \qquad \{f_x^\dagger, f_y\} = \delta_{x,y}. \tag{8}$$

This mapping represents the main technical tool of our analysis: In this section we will pedagogically review the three main steps of the mapping, while in Sec. 4 we will use it to determine the real-time evolution of interesting observables. Note that Kumar's mapping has originally been developed in the case of open boundary conditions but, by addressing some additional complications, it has later been extended to the periodic case [90]. Here we stick to the open-chain case for the sake of simplicity.

## 3.1 Operatorial Mapping

The first step is to define a formal operatorial mapping between the set of spinful fermions $\{c_{x,a}^\dagger\}_{a=\uparrow,\downarrow}$ and a set $\{f_x, \sigma_{a,x}\}_{a=1,2,3}$ composed by canonical spinless fermions $f_x^\dagger$ fulfilling (8) and Pauli matrices $\{\sigma_{a,x}\}_{a=1,2,3}$. In particular, considering a chain with an even number of sites $L$ we define

$$c_{x,\uparrow}^\dagger = a_x^x \sigma_{+,x}, \qquad\qquad c_{x,\downarrow}^\dagger = \tfrac{1}{2}(ia_x^y - a_x^x \sigma_{3,x}), \qquad\qquad x \text{ odd} \tag{9a}$$

$$c_{x,\uparrow}^\dagger = ia_x^y \sigma_{+,x}, \qquad\qquad c_{x,\downarrow}^\dagger = \tfrac{1}{2}(a_x^x - ia_x^y \sigma_{3,x}), \qquad\qquad x \text{ even} \tag{9b}$$

where we introduced the Majorana fermions

$$a_x^x = f_x^\dagger + f_x, \qquad\qquad ia_x^y = f_x^\dagger - f_x, \tag{10}$$

and the spin-ladder operators

$$\sigma_{\pm,x} = \frac{\sigma_{1,x} \pm i\sigma_{2,x}}{2}. \tag{11}$$

Under the mapping (9) the states (5) transform as follows

$$|0\rangle \mapsto |\bigcirc\rangle |-\rangle, \qquad\qquad c_{x,\uparrow}^\dagger |0\rangle \mapsto f_x^\dagger \sigma_{+,x} |\bigcirc\rangle |-\rangle, \tag{12}$$

$$c_{x,\downarrow}^\dagger |0\rangle \mapsto f_x^\dagger |\bigcirc\rangle |-\rangle, \qquad\qquad c_{x,\uparrow}^\dagger c_{x,\downarrow}^\dagger |0\rangle \mapsto (-1)^{x+1} \sigma_{+,x} |\bigcirc\rangle |-\rangle, \tag{13}$$

where we respectively denoted by $|\bigcirc\rangle$ and $|-\rangle$ the spinless-fermion vacuum and the spin-down state. These states are defined by

$$f_x |\bigcirc\rangle = 0, \qquad\qquad \sigma_{-,x} |-\rangle = 0, \qquad\qquad \forall x. \tag{14}$$

The explicit form of the inverse of (9) depends on the parity of $x$. In particular, for odd $x$ we have

$$f_x^\dagger = (1 - n_{x,\uparrow})c_{x,\downarrow}^\dagger - n_{x,\uparrow}c_{x,\downarrow}, \qquad \sigma_{3,x} = 2n_{x,\uparrow} - 1, \qquad \sigma_{+,x} = c_{x,\uparrow}^\dagger(c_{x,\downarrow}^\dagger + c_{x,\downarrow}), \tag{15a}$$

$$a_x^x = (1 - 2n_{x,\uparrow})(c_{x,\downarrow}^\dagger + c_{x,\downarrow}), \qquad ia_x^y = c_{x,\downarrow}^\dagger - c_{x,\downarrow}, \tag{15b}$$

while for even $x$ we find

$$f_x^\dagger = n_{x,\uparrow}c_{x,\downarrow}^\dagger + (1 - n_{x,\uparrow})c_{x,\downarrow}, \qquad \sigma_{3,x} = 2n_{x,\uparrow} - 1, \qquad \sigma_{+,x} = c_{x,\uparrow}^\dagger(c_{x,\downarrow}^\dagger - c_{x,\downarrow}), \tag{15c}$$

$$a_x^x = c_{x,\downarrow}^\dagger + c_{x,\downarrow}, \qquad\qquad ia_x^y = (1 - 2n_{x,\uparrow})(c_{x,\downarrow}^\dagger - c_{x,\downarrow}). \tag{15d}$$

For future reference we also note the following simple relationship between the local number operators in the spinful and spinless fermions

$$\left(n_{x,\uparrow} - \frac{1}{2}\right)\left(n_{x,\downarrow} - \frac{1}{2}\right) = \frac{1}{2}\left(\frac{1}{2} - d_x\right), \tag{16}$$

where we introduced the local number operator for spinless fermions

$$d_x \equiv f_x^\dagger f_x \, . \tag{17}$$

Finally, applying the mapping (9) to the Hamiltonian (1) one finds

$$H = -J \sum_{x=1}^{L-1} \left[ (f_{x+1}^\dagger f_x + f_x^\dagger f_{x+1}) X_{x,x+1} + (-1)^x (f_{x+1}^\dagger f_x^\dagger + f_x f_{x+1})(X_{x,x+1} - 1) \right]$$
$$- \frac{U}{2} \sum_{x=1}^{L} \left( f_x^\dagger f_x - \frac{1}{2} \right), \tag{18}$$

where we introduced the SWAP operator

$$X_{x,x+1} = \frac{1}{2} + \frac{1}{2} \sum_{a=1,2,3} \sigma_{a,x+1} \sigma_{a,x} \, , \tag{19}$$

which exchanges the spin states at positions $x$ and $x + 1$. For the sake of completeness we report the straightforward calculation in Appendix A.

## 3.2 Infinite-repulsion Limit

The form (18) of the Hamiltonian makes it very simple to understand the effect of the $U \to \infty$ limit. Indeed, $U$ appears in (18) only as a chemical potential for the spinless fermions. This means that the sectors of the Hilbert space with fixed number of fermions are separated by an energy gap proportional to $U$ and changing the particle number costs increasingly more energy for increasing values of $U$, becoming infinitely expensive in the limit $U \to \infty$. As a result, the effect of the limit is to fix the particle number.

As we now review (see also Ref. [89]), the standard way to formalise this intuition is to employ a Schrieffer-Wolff transformation to the Hamiltonian $H/U$. The idea is to apply a unitary transformation of the form

$$W = e^{iS} \, . \tag{20}$$

Here $S$ is an Hermitian operator with a regular expansion in $J/U$, which is chosen to explicitly move the terms that do not conserve the number of spinless fermions to higher orders in $J/U$. Let us begin by breaking the Hamiltonian into four parts

$$H = UD + JT_0 + JT_2 + JT_{-2} \, , \tag{21}$$

where

$$D = \frac{L}{4} - \frac{1}{2} \sum_{x=1}^{L} f_x^\dagger f_x \, , \tag{22a}$$

$$T_0 = -J \sum_{x=1}^{L-1} (f_{x+1}^\dagger f_x + f_x^\dagger f_{x+1}) X_{x,x+1} \, , \tag{22b}$$

$$T_2 = T_{-2}^\dagger = -J \sum_{x=1}^{L-1} (-1)^x f_{x+1}^\dagger f_x^\dagger (X_{x,x+1} - 1) \, . \tag{22c}$$

The operators $D$ and $T$ conserve the number of spinless fermions, and the operators $T_2$ and $T_{-2}$ increase and decrease their number by two, respectively. These operators satisfy the following commutation relations

$$[T_n, D] = \frac{n}{2} T_n \, , \qquad n = \pm 2 \, . \tag{23}$$

Next we determine $S$ such that the transformation (20) moves the operators $T_2$ and $T_{-2}$ to higher order terms in $J/U$. We define the transformed Hamiltonian as

$$H' = e^{iS} H e^{-iS} = H + [iS, H] + \tfrac{1}{2}[iS, [iS, H]] + \dots \tag{24}$$

and take $S$ to have a regular power series expansion of the form

$$S = \frac{J}{U} S^{(0)} + \frac{J^2}{U^2} S^{(1)} + \frac{J^3}{U^3} S^{(2)} + \dots . \tag{25}$$

Performing the transformation, we substitute (21) and (25) into (24) and divide by $U$

$$
\begin{aligned}
\frac{H'}{U} = {}& D + \frac{J}{U}\left(T_0 + T_2 + T_{-2} + [iS^{(0)}, D]\right) \\
& + \frac{J^2}{U^2}\left([iS^{(0)}, (T_0 + T_2 + T_{-2})] + \frac{1}{2}\left[iS^{(0)}, [iS^{(0)}, D]\right] + [iS^{(1)}, D]\right) + \dots .
\end{aligned}
\tag{26}
$$

To remove the terms that do not conserve the number of fermions at order $J/U$, we require

$$T_2 + T_{-2} + [iS^{(0)}, D] = 0. \tag{27}$$

Taking

$$iS^{(0)} = -T_2 = T_{-2}, \tag{28}$$

satisfies this condition and simplifies the transformed Hamiltonian to

$$\frac{H'}{U} = D + \frac{J}{U} T_0 + \mathcal{O}\left(\frac{J^2}{U^2}\right). \tag{29}$$

This procedure can be continued, but for our purposes it is sufficient to stop here, giving a transformed Hamiltonian of the form

$$H' = \frac{UL}{4} - \frac{U}{2}\sum_{x=1}^{L} f_x^\dagger f_x - J\sum_{x=1}^{L-1}(f_{x+1}^\dagger f_x + f_x^\dagger f_{x+1})X_{x,x+1} + \mathcal{O}\left(\frac{J^2}{U}\right). \tag{30}$$

To take the strong-coupling limit, we act on states with a fixed number of particles, so that the first two terms contribute a constant which we can ignore. Therefore, in the limit of infinite interaction the Hamiltonian becomes

$$H_\infty = -J\sum_{x=1}^{L-1} X_{x,x+1}(f_{x+1}^\dagger f_x + f_x^\dagger f_{x+1}), \tag{31}$$

which, as promised, conserves the number of spinless fermions.

## 3.3 Kumar's Transformation

The final step of Kumar's mapping is to remove the spin terms $X_{x,x+1}$ from (31). This can be done by applying the following unitary transformation [86]

$$\mathcal{U} \equiv U_2 U_3 \cdots U_L, \qquad U_x \equiv (1 - d_x) + d_x \mathcal{X}_x, \tag{32}$$

where

$$\mathcal{X}_x \equiv X_{x,x-1} X_{x-1,x-2} \dots X_{2,1}, \tag{33}$$

is the operator implementing a periodic one-site shift to the left in a spin-1/2 chain of $x$ sites. Note that the operators $X_{x,x+1} = X_{x+1,x}$ are both unitary and Hermitian, whereas both $\mathcal{X}_x$ and $U_x$ are unitary but not Hermitian

$$X_{x,x-1}^2 = I, \qquad \mathcal{X}_x^\dagger \mathcal{X}_x = I, \qquad U_x^\dagger U_x = I, \tag{34}$$

where $I$ represents the identity operator. Moreover

$$[O_x, U_y] = 0, \qquad y < x, \tag{35}$$

where $O_x$ is a generic operator acting non-trivially only at $x$.

Let us now act with $\mathcal{U}$ on a generic single term in the sum of the Hamiltonian (31), specifically

$$X_{x+1,x}(f_{x+1}^\dagger f_x + f_x^\dagger f_{x+1}). \tag{36}$$

The first $x-1$ terms in the product $\mathcal{U}$ act trivially, leaving the expression unchanged. This can be seen by using the commutation relations above to commute all the $U_x$ operators through to their daggered counterparts where they annihilate each other

$$U_{x-1}^\dagger U_{x-2}^\dagger \ldots U_2^\dagger X_{x+1,x}(f_{x+1}^\dagger f_x + f_x^\dagger f_{x+1})U_2 \ldots U_{x-2}U_{x-1} = X_{x+1,x}(f_{x+1}^\dagger f_x + f_x^\dagger f_{x+1}). \tag{37}$$

The next term in the product $\mathcal{U}$ acts non-trivially. In particular we have

$$\begin{aligned} U_x^\dagger X_{x+1,x}(f_{x+1}^\dagger f_x + f_x^\dagger f_{x+1})U_x &= U_x^\dagger X_{x+1,x}(f_{x+1}^\dagger f_x \mathcal{X}_{x,x-1} + f_x^\dagger f_{x+1}) \\ &= f_{x+1}^\dagger f_x X_{x+1,x} \mathcal{X}_x + \mathcal{X}_x^{-1} X_{x+1,x} f_x^\dagger f_{x+1} \\ &= f_{x+1}^\dagger f_x \mathcal{X}_{x+1} + \mathcal{X}_{x+1}^{-1} f_x^\dagger f_{x+1}. \end{aligned} \tag{38}$$

In the first line we acted with the fermions on the operator $U_x$: since the first product of two fermions has an annihilation operator at site $x$, it picks up a factor of $\mathcal{X}_x$. The second term has a creation operator at site $x$ and therefore remains unchanged. In the second line, we applied the operator $U_x^\dagger$, which results in the second term gaining a factor of $\mathcal{X}_x^\dagger$. In the last line we collected the $X_{x+1,x}$ into the product $X$ operators denoted by $\mathcal{X}$.

Acting with the next term in the product $\mathcal{U}$ serves to remove the spin terms entirely

$$\begin{aligned} U_{x+1}^\dagger(f_{x+1}^\dagger f_x \mathcal{X}_{x+1} + \mathcal{X}_{x+1}^\dagger f_x^\dagger f_{x+1})U_{x+1} &= f_{x+1}^\dagger f_x \mathcal{X}_{x+1}^{-1} \mathcal{X}_{x+1} + \mathcal{X}_{x+1}^{-1} \mathcal{X}_{x+1} f_x^\dagger f_{x+1} \\ &= f_{x+1}^\dagger f_x + f_x^\dagger f_{x+1}. \end{aligned} \tag{39}$$

Finally, the remaining terms act trivially, since they commute with the fermions and are unitary

$$U_L^\dagger U_{L-1}^\dagger \ldots U_{x+2}^\dagger(f_{x+1}^\dagger f_x + f_x^\dagger f_{x+1})U_{x+2} \ldots U_{L-1}U_L = f_{x+1}^\dagger f_x + f_x^\dagger f_{x+1}. \tag{40}$$

In summary, we have shown that

$$\mathcal{U}^\dagger X_{x+1,x}(f_{x+1}^\dagger f_x + f_x^\dagger f_{x+1})\mathcal{U} = f_{x+1}^\dagger f_x + f_x^\dagger f_{x+1}, \tag{41}$$

which leads immediately to Kumar's result [86]

$$H_{\text{tb}} = \mathcal{U}^\dagger H_\infty \mathcal{U} = -J \sum_{x=1}^{L-1}(f_{x+1}^\dagger f_x + f_x^\dagger f_{x+1}). \tag{42}$$

This free Hamiltonian can now be readily diagonalised using the standard Fourier transform for open boundary conditions

$$f_x = \sqrt{\frac{2}{L+1}} \sum_k f(k)\sin(kx), \qquad k = \frac{n\pi}{L+1}, \quad n = 1, 2, \ldots, L, \tag{43}$$

resulting in

$$H_{\text{tb}} = \sum_k \varepsilon(k) f^\dagger(k) f(k), \qquad \varepsilon(k) = -2J \cos k. \tag{44}$$

Before concluding this survey we note that the transformation (32) is somewhat "asymmetric" as it treats the two boundaries of the system differently. Indeed, the operator $U_x$ acts non-trivially only on $[1, x]$. As might be expected, one can also use an alternative unitary transformation to map (31) into (42) which acts "from the other edge" of the system, namely

$$\mathcal{V} \equiv V_{L-1} V_{L-2} \cdots V_1, \qquad V_x \equiv (1 - d_x) + d_x \mathcal{X}_x \mathcal{X}_L^{-1}, \tag{45}$$

where

$$\mathcal{X}_x \mathcal{X}_L^{-1} = X_{x+1,x} X_{x+2,x+1} \ldots X_{L,L-1}, \tag{46}$$

act non-trivially only on $[x, L]$. Proceeding as above one can readily show

$$H_{\text{tb}} \equiv \mathcal{V}^\dagger H_\infty \mathcal{V}. \tag{47}$$

Although (32) and (45) give equivalent results, in the following we will use the original formulation (32) as it is somewhat more intuitive.

## 4 Time evolution of local observables

The fact that the Hubbard model for infinite repulsion can be mapped to a free Hamiltonian opens the possibility to directly compute the time evolution of local observables after a quantum quench. The task then is to choose local operators and initial states that transform simply under the Kumar transformation. In this section we fix a family of simple initial states and present two classes of operators which turn out to have accessible dynamics. Specifically, we consider quenches from generic product states of the form

$$|\Psi\rangle = \prod_{j=1}^N c_{x_j, \sigma_j}^\dagger |0\rangle, \qquad x_j \in \{1, \ldots, L\}, \quad \sigma_j = \uparrow, \downarrow, \tag{48}$$

where $N \leq L$. The key property of (48) is that it takes a factorised form in terms of the target operators of the Kumar mapping. Namely, under the mapping (9) it becomes

$$|\Psi\rangle = \prod_{j=1}^N f_{x_j}^\dagger \prod_{j=1}^N (\delta_{\sigma_j, \uparrow} \sigma_{+, x_j} + \delta_{\sigma_j, \downarrow} I) |\bigcirc\rangle |-\rangle. \tag{49}$$

We now proceed to identify two classes of "solvable" operators.

### 4.1 Analytic Functions of the Total Number Operator

The first class of operators with solvable dynamics are $F(n_{x_1}, \ldots, n_{x_m})$ where $F(z_1, \ldots, z_m)$ is an arbitrary analytic function of $m$ variables and

$$n_x \equiv n_{x, \uparrow} + n_{x, \downarrow}, \tag{50}$$

is the operator counting the number of spin-full fermions at position $x$. Indeed, we have the following property

**Property 1.** *For any analytic function $F(z_1, \ldots, z_m)$ we have*

$$\langle\Psi|e^{itH_\infty}F(n_{x_1},\ldots,n_{x_m})e^{-itH_\infty}|\Psi\rangle = \langle\Psi_f|e^{itH_{tb}}F(d_{x_1},\ldots,d_{x_m})e^{-itH_{tb}}|\Psi_f\rangle\,, \qquad (51)$$

*where $H_{tb}$ is the tight binding Hamiltonian (cf. (42)), $d_x$ is the local number operator of spinless fermions (cf. (17)), and we defined*

$$|\Psi_f\rangle = \prod_{j=1}^{N} f_{x_j}^\dagger |\bigcirc\rangle\,. \qquad (52)$$

*Proof.* We prove the property in two steps. First we show

$$\langle\Psi|e^{itH_\infty}F(n_{x_1},\ldots,n_{x_m})e^{-itH_\infty}|\Psi\rangle = \langle\Psi|e^{itH_\infty}F(d_{x_1},\ldots,d_{x_m})e^{-itH_\infty}|\Psi\rangle\,, \qquad (53)$$

and then

$$\langle\Psi|e^{itH_\infty}F(d_{x_1},\ldots,d_{x_m})e^{-itH_\infty}|\Psi\rangle = \langle\Psi_f|e^{itH_{tb}}F(d_{x_1},\ldots,d_{x_m})e^{-itH_{tb}}|\Psi_f\rangle\,. \qquad (54)$$

Since $F(z_1, \ldots, z_m)$ is analytic we can establish the first step considering generic monomials of the form

$$\langle\Psi|e^{itH_\infty}n_{x_1}^{p_1}\cdots n_{x_m}^{p_m}e^{-itH_\infty}|\Psi\rangle\,. \qquad (55)$$

Now we use (16) and (9) to obtain

$$n_x = n_{x,\uparrow} + n_{x,\downarrow} = d_x - 2n_{x,\uparrow}n_{x,\downarrow}$$
$$= d_x - 2\left(\frac{1+\sigma_{3,x}}{2}\right)\left(\frac{1-\sigma_{3,x}(2d_x-1)}{2}\right)\,. \qquad (56)$$

In words this means that $n_x$ and $d_x$ only differ by the operator counting the number of double occupancies. Therefore, we have

$$n_{x_m}^{p_m}e^{-itH_\infty}|\Psi\rangle = d_{x_m}^{p_m}e^{-itH_\infty}|\Psi\rangle\,. \qquad (57)$$

Indeed, the state (48) does not have double occupancies and the latter are not produced during the time evolution. This is obvious considering the form (6) of the Hamiltonian but can also be seen from (49) and (31). Indeed, when expressed in terms of spinless fermions and Pauli spins, the state $|\Psi\rangle$ has up spins only in positions occupied by spinless fermions and this property is preserved by the Hamiltonian (31). Therefore, the second operator on the second line of (56) annihilates it at all times.

Using (57) and the fact that $n_x$ commute for different $x$ we immediately have (53). To prove the second step we perform the Kumar rotation (32). Namely

$$\langle\Psi|e^{itH_\infty}F(d_{x_1},\ldots,d_{x_m})e^{-itH_\infty}|\Psi\rangle = \langle\Psi|\mathcal{U}\mathcal{U}^\dagger e^{itH_\infty}F(d_{x_1},\ldots,d_{x_m})e^{-itH_\infty}\mathcal{U}\mathcal{U}^\dagger|\Psi\rangle$$
$$= \langle\Psi|\mathcal{U}e^{itH_{tb}}F(d_{x_1},\ldots,d_{x_m})e^{-itH_{tb}}\mathcal{U}^\dagger|\Psi\rangle\,, \qquad (58)$$

where we used that $\mathcal{U}^\dagger d_x\mathcal{U} = d_x$. To conclude we explicitly evaluate the action of $\mathcal{U}^\dagger$ on the state $|\Psi\rangle$

$$\mathcal{U}^\dagger|\Psi\rangle = \left((1-n_L)+n_L\mathcal{X}_L^\dagger\right)\cdots\left((1-n_2)+n_2\mathcal{X}_2^\dagger\right)\prod_{j=1}^{N}f_{x_j}^\dagger\prod_{j=1}^{N}(\delta_{\sigma_j,\uparrow}\sigma_{+,x_j}+\delta_{\sigma_j,\downarrow}I)|\bigcirc\rangle|-\rangle$$

$$= |\Psi_f\rangle\left(\mathcal{X}_{x_N}^\dagger\mathcal{X}_{x_{N-1}}^\dagger\cdots\mathcal{X}_{x_2}^\dagger\mathcal{X}_{x_1}^\dagger\prod_{j=1}^{N}(\delta_{\sigma_j,\uparrow}\sigma_{+,x_j}+\delta_{\sigma_j,\downarrow}I)|-\rangle\right)$$

$$= |\Psi_f\rangle\underbrace{|\sigma_N,\sigma_{N-1},\ldots,\sigma_1,}_{N}\underbrace{-,\ldots,-\rangle}_{L-N}\,, \qquad (59)$$

where we set $\mathcal{X}_1 = I$. Since the transformed version of the state maintains a product form and $F(d_{x_1},\ldots,d_{x_m})$ acts as the identity on the spin part we immediately have (54). $\qquad\square$

Expressed in physical terms Property 1 states that the evolution of any function of the density of spin-full fermions is mapped directly into that of the density of spin-less fermions. In other words, from the point of view of this class of observables the Hubbard model for $U = \infty$ behaves as a single gas of free fermions. This should be compared with the $U = 0$ limit, where, instead, it corresponds to two independent free fermionic gases.

On the practical level this means that the time-evolution of the expectation value of $F(n_{x_1}, \ldots, n_{x_m})$ can can be easily computed using free fermion techniques. For instance, considering a state $|\Psi_N\rangle$ with Néel ordering for the particles but with arbitrary spin configuration, i.e.

$$|\Psi_N\rangle = \prod_{x=1}^{L/2} c^\dagger_{2x,\sigma_{2x}} |0\rangle \,, \qquad \sigma_{2x} = \uparrow, \downarrow \,, \tag{60}$$

the one- and two- point functions of the density operator are explicitly evaluated as

$$C_{\Psi_N}(x,x,t) = \frac{1}{2(L+1)} \sum_k \left(1 + (-1)^x e^{-4itJ\cos(k)}\right)\left(1 - \cos(2kx)\right),$$

$$D_{\Psi_N}(x,y,t) = \frac{1}{(L+1)^2} \sum_{k_1,k_2} (1 + (-1)^y e^{-4itJ\cos(k_1)})(1 - (-1)^y e^{4itJ\cos(k_2)})$$
$$\times \sin(k_1 x)\sin(k_1 y)\sin(k_2 x)\sin(k_2 y), \tag{61}$$

where we introduced fermionic and density-density two-point correlators

$$C_\Psi(x,y,t) \equiv \langle\Psi|e^{itH_\infty} c^\dagger_{x,\uparrow} c_{y,\uparrow} e^{-itH_\infty}|\Psi\rangle + \langle\Psi|e^{itH_\infty} c^\dagger_{x,\downarrow} c_{y,\downarrow} e^{-itH_\infty}|\Psi\rangle \,, \tag{62}$$

$$D_\Psi(x,y,t) \equiv \langle\Psi|e^{itH_\infty} n_x n_y e^{-itH_\infty}|\Psi\rangle - \langle\Psi|e^{itH_\infty} n_x e^{-itH_\infty}|\Psi\rangle\langle\Psi|e^{itH_\infty} n_y e^{-itH_\infty}|\Psi\rangle \,.$$

Note that $C_\Psi(x,x,t)$ only depends on $d_x$ even though $C_\Psi(x,y,t)$ for $x \neq y$ does not.

In the thermodynamic limit $L \to \infty$ the above expectation values can be written as

$$C_{\Psi_N}(x,x,t)_{\text{th}} = \lim_{\text{th}} C_{\Psi_N}(x,x,t) = \frac{1}{2}\left[1 + (-1)^x J_0(4Jt) - J_{2x}(4Jt)\right]. \tag{63}$$

$$D_{\Psi_N}(x,y,t)_{\text{th}} = \lim_{\text{th}} D_{\Psi_N}(x,y,t) = \delta_{x,y} - \left|J_{x-y}(4Jt) + (-1)^{y+1} J_{x+y}(4Jt)\right|^2. \tag{64}$$

Where we introduced the Bessel functions of the first kind

$$J_n(z) = \frac{1}{2\pi} \int_{-\pi}^\pi dk \, e^{i(nk - z\sin(k))}. \tag{65}$$

The dynamics described by Eqs. (63) and (64) are illustrated in Fig. (1).

We remark that Property 1 can also be used to compute more complicated functions of $\{n_x\}_{x=1}^L$. An interesting example is the time-evolution of the full counting statistics of

$$N_A = \sum_{y \in A} n_y \,, \tag{66}$$

i.e., the total number of spinfull fermions in a given subsystem $A \subset \{1, \ldots, L\}$, on the time evolving state $e^{-itH_\infty} |N\rangle$. The full counting statistics of a certain observable $\mathcal{O}$ in a state $|\Phi\rangle$ is the probability distribution for the outcomes of measurements of $\mathcal{O}$ in $|\Phi\rangle$ and, as such, encodes detailed information about the quantum fluctuations $\mathcal{O}$. In particular, over the last few years there has been increasing interest in the time-evolution of full counting statistics after quantum quenches [91–97].

The full counting statistics is completely specified by its Fourier transform, the characteristic function of the probability distribution, which in our case reads as

$$\chi_\Psi(t, A, \lambda) \equiv \langle\Psi|e^{itH_\infty} e^{i\lambda N_A} e^{-itH_\infty}|\Psi\rangle = \langle\Psi_f|e^{itH_{\text{tb}}} e^{i\lambda D_A} e^{-itH_{\text{tb}}}|\Psi_f\rangle \,, \tag{67}$$

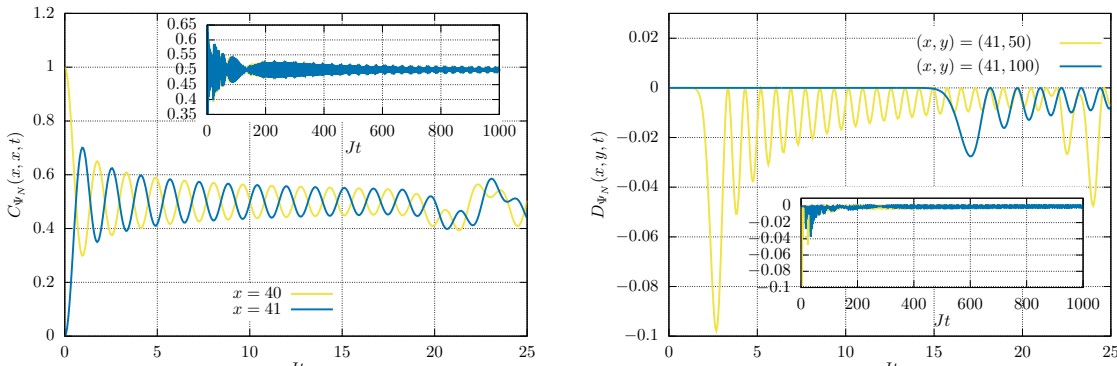

Figure 1: Time evolution of one-point function (left) and connected two-point function (right) of the total density operator $n_x$ after a quench from an initial state with Néel order in the particles (cf. (60)) in the thermodynamic limit (the insets report the long time behaviour). The one-point function relaxes to $1/2$ in the infinite-time limit while the connected two-point function goes to $\delta_{x,0}$. Note that at times $Jt = 2x/4$ and $Jt = (x+y)/4$ the correlations begin to be affected by the boundary (the maximal velocity for the propagation of correlations is $v_{\max} = 4J$).

where we again used Property 1 and we introduced

$$D_A = \sum_{y \in A} d_y. \tag{68}$$

The r.h.s. of (67) is immediately expressed as the determinant of a $|A| \times |A|$ matrix [93, 100, 101]. In particular we have

$$\chi_\Psi(t, A, \lambda) = \det(I + (e^{i\lambda} - 1)\mathbb{C}_A^{\Psi_f}(t)), \tag{69}$$

where $I$ denotes the identity matrix and

$$[\mathbb{C}_A^{\Psi_f}(t)]_{x,y} = \langle \Psi_f | e^{itH_{\text{tb}}} f_x^\dagger f_y e^{-itH_{\text{tb}}} | \Psi_f \rangle, \qquad x, y = 1, \dots, |A|, \tag{70}$$

is the correlation matrix reduced to the subsystem $A$. In particular, for a state with Néel order in the particles (cf. (60)) we have

$$[\mathbb{C}_A^{N_f}(t)]_{x,y} = \delta_{x,y} + \frac{(-1)^y}{L+1} \sum_k \sin(kx)\sin(ky)e^{-4iJ\cos(k)t}, \tag{71}$$

where we introduced

$$|N_f\rangle = \prod_{x=0}^{L/2-1} f_{2x+1}^\dagger |\bigcirc\rangle. \tag{72}$$

Specifically, in the thermodynamic limit we find

$$\lim_{\text{th}} [\mathbb{C}_A^{N_f}(t)]_{x,y} = \delta_{x,y} - \frac{i^{x+y}}{2}(J_{x-y}(4Jt) - (-1)^x J_{x+y}(4Jt)), \qquad x, y = 1, \dots, |A|. \tag{73}$$

The time evolution of the characteristic function (67) for two different values of $\lambda$ is depicted in Fig. 2. We see that the Néel order melts very rapidly after the quench.

Before concluding this subsection we note that Property 1 can be immediately extended to initial states of the form

$$|\Psi^{(g)}\rangle = \left[ \sum_{\{y_j\},\{s_j\}} A_{\{y_j\},\{s_j\}} \prod_{j=1}^N c_{y_j,\sigma_j}^\dagger \right] |0\rangle, \tag{74}$$

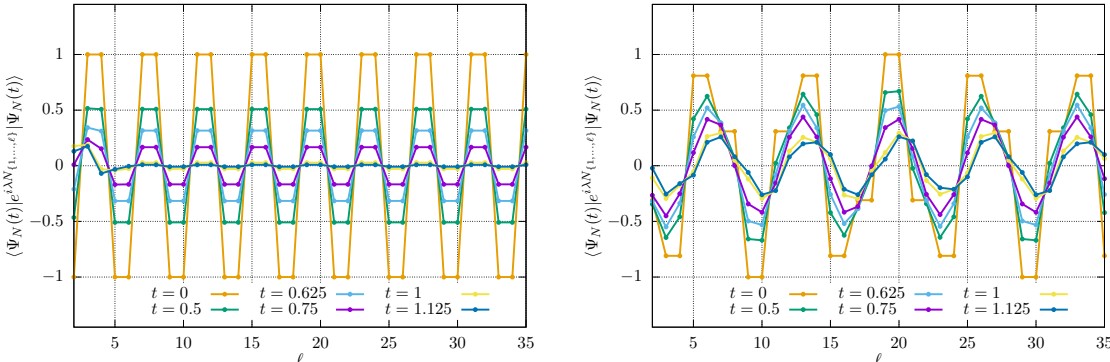

Figure 2: Characteristic function $\chi_N(t,A,\lambda)$ (cf. (67)) vs subsystem size for different times after a quench from the state (60) in the thermodynamic limit. The two panels correspond to $\lambda = \pi$ (left) and $\lambda = 7\pi/5$ (right). In both cases $\chi_N(t,A,\lambda)$ approaches 0 for all $\ell$ for large times, signalling the melting of the order.

where $|\{y_j\}| = |\{s_j\}| = N < L$. In this case, using that $d_x$ acts diagonally on the spin sector we find

$$\langle \Psi^{(g)} | e^{itH_\infty} F(n_{x_1}, \dots, n_{x_m}) e^{-itH_\infty} | \Psi^{(g)} \rangle = \tag{75}$$
$$= \sum_{\{y_j\},\{s_j\}} |A_{\{y_j\},\{s_j\}}|^2 \langle \Psi_{f\{y_j\}} | e^{itH_{tb}} F(d_{x_1}, \dots, d_{x_m}) e^{-itH_{tb}} | \Psi_{f\{y_j\}} \rangle \,.$$

Here, for the sake of clarity, we reported the explicit dependence of the state (52) on the set of particle positions $\{y_j\}$.

## 4.2 Two-point correlators

The second class of operators whose dynamics simplifies when evolving from the states (48) are

$$c^{(\dagger)}_{x,\uparrow(\downarrow)} c^{(\dagger)}_{y,\uparrow(\downarrow)}, \tag{76}$$

i.e., quadratic monomials of spin-full operators. In particular, we consider the only two equal-time two point functions which are not identically zero because violating spin or particle number conservation, namely

$$C_{\uparrow,\Psi}(x,y,t) \equiv \langle \Psi | e^{itH_\infty} c^\dagger_{x,\uparrow} c_{y,\uparrow} e^{-itH_\infty} | \Psi \rangle, \tag{77}$$

$$C_{\downarrow,\Psi}(x,y,t) \equiv \langle \Psi | e^{itH_\infty} c^\dagger_{x,\downarrow} c_{y,\downarrow} e^{-itH_\infty} | \Psi \rangle. \tag{78}$$

Firstly, we use a simple argument to rewrite the second correlator as the expectation of the operator with both spins up, with respect to a spin-flipped state. Let $\mathcal{F}$ be the unitary spin-flip operator such that

$$\mathcal{F} c^{(\dagger)}_{x,\uparrow} \mathcal{F}^\dagger = c^{(\dagger)}_{x,\downarrow}, \qquad \mathcal{F} c^{(\dagger)}_{x,\downarrow} \mathcal{F}^\dagger = c^{(\dagger)}_{x,\uparrow}. \tag{79}$$

Inserting this operator into the correlator gives

$$C_{\downarrow,\Psi}(x,y,t) = \langle \Psi | \mathcal{F}^\dagger \mathcal{F} e^{itH_\infty} \mathcal{F}^\dagger \mathcal{F} c^\dagger_{x,\downarrow} \mathcal{F}^\dagger \mathcal{F} c_{y,\downarrow} \mathcal{F}^\dagger \mathcal{F} e^{-itH_\infty} \mathcal{F}^\dagger \mathcal{F} | \Psi \rangle \,. \tag{80}$$

Noting that Hamiltonian (1) is invariant under this transformation, the expectation value can be rewritten as

$$C_{\downarrow,\Psi}(x,y,t) = C_{\uparrow,\Psi'}(x,y,t), \tag{81}$$

where

$$|\Psi'\rangle = \prod_{j=1}^{N} c_{x_j,\bar{\sigma}_j}^\dagger |0\rangle , \qquad \bar{\uparrow} = \downarrow , \qquad \bar{\downarrow} = \uparrow . \tag{82}$$

Therefore, to calculate the correlators of interest, we will calculate the expectation value of the operator with both spins up, but with respect to a generic state $|\Psi\rangle$. This makes the calculation easier, as the Kumar mapping (9) is simpler on fermions with spin up.

Employing a similar reasoning we can also restrict ourselves to the case $\max(x,y) > L/2$, which, as we will see, is easier to treat with the unitary transformation (45). Indeed, if $\max(x,y) \leq L/2$ we can apply the unitary reflection operator $\mathcal{R}$ acting as

$$\mathcal{R} c_{x,\uparrow}^{(\dagger)} \mathcal{R}^\dagger = c_{L+1-x,\uparrow}^{(\dagger)}, \qquad \mathcal{R} c_{x,\downarrow}^{(\dagger)} \mathcal{R}^\dagger = c_{L+1-x,\downarrow}^{(\dagger)}. \tag{83}$$

This gives

$$C_{\uparrow,\Psi}(x,y,t) = C_{\uparrow,\Psi''}(L+1-x, L+1-y, t), \tag{84}$$

with

$$|\Psi''\rangle = \prod_{j=1}^{N} c_{L+1-x_j,\sigma_j}^\dagger |0\rangle . \tag{85}$$

Analogously

$$C_{\downarrow,\Psi}(x,y,t) = C_{\uparrow,\Psi'''}(L+1-x, L+1-y, t), \tag{86}$$

with

$$|\Psi'''\rangle = \prod_{j=0}^{N} c_{L+1-x_j,\bar{\sigma}_j}^\dagger |0\rangle . \tag{87}$$

In summary, to evaluate $C_{\uparrow,\Psi}(x,y,t)$ and $C_{\downarrow,\Psi}(x,y,t)$ for all $x$ and $y$ here we consider

$$C_{\uparrow,\Psi}(x,y,t), \qquad \max(x,y) > L/2, \tag{88}$$

and access all the other cases by performing the appropriate spin-flip and reflections of the state. We begin our analysis of $C_{\uparrow,\Psi}(x,y,t)$: applying the Kumar mapping, the correlators are written as

$$C_{\uparrow,\Psi}(x,y,t) = \langle\Psi|e^{itH\infty}(f_x^\dagger + (-1)^{x+1} f_x)(f_y + (-1)^{y+1} f_y^\dagger)\sigma_{+,x}\sigma_{-,y} e^{-itH\infty}|\Psi\rangle , \tag{89}$$

where the Hamiltonian is given by (18). This can be further simplified using the fact that the initial state has fixed particle number

$$C_{\uparrow,\Psi}(x,y,t) = \langle\Psi|e^{itH\infty}(f_x^\dagger f_y + (-1)^{x+y} f_x f_y^\dagger)\sigma_{+,x}\sigma_{-,y} e^{-itH\infty}|\Psi\rangle . \tag{90}$$

Next we apply the unitary transformation (32) to write the infinite interaction Hamiltonian as the free Hamiltonian $H_{\text{tb}}$ (42) in the spinless fermions, i.e.

$$C_{\uparrow,\Psi}(x,y,t) = \langle\Psi|\mathcal{U}\mathcal{U}^\dagger e^{itH\infty}\mathcal{U}\mathcal{U}^\dagger(f_x^\dagger f_y + (-1)^{x+y} f_x f_y^\dagger)\sigma_{+,x}\sigma_{-,y}\mathcal{U}\mathcal{U}^\dagger e^{-itH\infty}\mathcal{U}\mathcal{U}^\dagger|\Psi\rangle \tag{91}$$

$$= \langle\Psi_f| \langle\Psi_s| e^{itH_{\text{tb}}}\mathcal{U}^\dagger(f_x^\dagger f_y + (-1)^{x+y} f_x f_y^\dagger)\sigma_{+,x}\sigma_{-,y}\mathcal{U}e^{-itH_{\text{tb}}}|\Psi_f\rangle |\Psi_s\rangle . \tag{92}$$

Here we used

$$\mathcal{U}^\dagger |\Psi\rangle = |\Psi_f\rangle |\Psi_s\rangle , \tag{93}$$

with

$$|\Psi_f\rangle = \prod_{j=1}^{N} f_{x_j}^\dagger , \qquad |\Psi_s\rangle = |\underbrace{\sigma_N, \sigma_{N-1}, \ldots, \sigma_1}_{N}, \underbrace{-, \ldots, -}_{L-N}\rangle . \tag{94}$$

The unitary operator $\mathcal{U}$ is a product of operators $U_z$, and those with $z = 1, 2, \ldots, \min(x, y) - 1$ can be commuted left through the spins and spinless fermions, cancelling with their counterparts in the operator $\mathcal{U}^\dagger$. This leaves

$$C_{\uparrow,\Psi}(x, y, t) = \langle \Psi_f | \langle \Psi_s | e^{itH_{\mathrm{tb}}} U_L^\dagger \cdots U_{\min(x,y)}^\dagger (f_x^\dagger f_y + (-1)^{x+y} f_x f_y^\dagger)$$
$$\cdot \sigma_{+,x} \sigma_{-,y} U_{\min(x,y)} \cdots U_L e^{-itH_{\mathrm{tb}}} | \Psi_f \rangle | \Psi_s \rangle . \tag{95}$$

The operators $U_z$ act on the creation and annihilation operators in the fermions as follows

$$U_z^{(\dagger)} f_z^\dagger = \mathcal{X}_z^{(\dagger)}, \qquad U_z^{(\dagger)} f_z = I. \tag{96}$$

Therefore, noting that

$$[U_x^{(\dagger)}, f_y^{(\dagger)}] = 0, \qquad x \neq y, \tag{97}$$

we can replace $U_{x/y}$ in (95) with either $I$ or $\mathcal{X}_{x/y}$ depending on whether they are on the left or on the right of the fermionic operators. Specifically we find

$$C_{\uparrow,\Psi}(x, y, t) = \langle \Psi_t | T_\leftarrow \left[ \mathcal{X}_x^\dagger \prod_{z=\min(x,y)+1}^{\max(x,y)-1} U_z^\dagger \right] f_x^\dagger f_y \sigma_{+,x} \sigma_{-,y} T_\rightarrow \left[ \mathcal{X}_y \prod_{z=\min(x,y)+1}^{\max(x,y)-1} U_z^\dagger \right] | \Psi_t \rangle$$
$$+ (-1)^{x+y} \langle \Psi_t | T_\leftarrow \left[ \mathcal{X}_y^\dagger \prod_{z=\min(x,y)+1}^{\max(x,y)-1} U_z^\dagger \right] f_x f_y^\dagger \sigma_{+,x} \sigma_{-,y} T_\rightarrow \left[ \mathcal{X}_x \prod_{z=\min(x,y)+1}^{\max(x,y)-1} U_z^\dagger \right] | \Psi_t \rangle. \tag{98}$$

Here, to write this expression in compact form we introduced the following $z$-ordered products of non-commuting operators

$$T_\leftarrow \left[ O_x O_y' \right] = \begin{cases} O_x O_y' & x > y \\ O_y' O_x & x < y \end{cases}, \qquad T_\rightarrow \left[ O_x O_y' \right] = \begin{cases} O_y' O_x & x > y \\ O_x O_y' & x < y \end{cases}, \tag{99}$$

the state

$$| \Psi_t \rangle \equiv U_{\max(x,y)+1} \cdots U_L e^{-itH_{\mathrm{tb}}} | \Psi_f \rangle | \Psi_s \rangle, \tag{100}$$

and used the convention

$$\prod_{z=a}^{b} U_z^\dagger = I, \qquad \text{for} \qquad b < a. \tag{101}$$

To evaluate (98), we will next expand the operators $U_x$ which will then allow us to write each term as a product of a correlator in just the spinless fermions (depending on time) and a correlator in just the spins (independent of time). To express this succinctly we introduce the following set of projectors

$$\mathcal{P}_x^{(k)} \equiv \begin{cases} 1 - d_x, & k = 0 \\ d_x, & k = 1 \end{cases}. \tag{102}$$

In this succinct notation, the operators $U_x$ can be written as

$$U_x = \sum_{k=0,1} \mathcal{X}_x^k \mathcal{P}_x^{(k)}. \tag{103}$$

We can then write the two-point function as a linear combination of terms, where each term is the product of a correlator in the spinless fermions and a correlator in the spins. In particular

we have

$$C_{\uparrow,\Psi}(x,y,t) = \sum_{\{k_z\}\in\{0,1\}} \mathcal{C}_{\Psi_s}(x,y,\{k_z\}) \langle\Psi_f|e^{itH_{\text{tb}}}f_x^\dagger f_y \prod_{\substack{z=\min(x,y)+1\\z\neq\max(x,y)}}^{L} \mathcal{P}_z^{(k_z)}e^{-itH_{\text{tb}}}|\Psi_f\rangle$$

$$+ (-1)^{x+y}\sum_{\{k_z\}\in\{0,1\}} \mathcal{D}_{\Psi_s}(y,x,\{k_z\}) \langle\Psi_f|e^{itH_{\text{tb}}}f_x f_y^\dagger \prod_{\substack{z=\min(x,y)+1\\z\neq\max(x,y)}}^{L} \mathcal{P}_z^{(k_z)}e^{-itH_{\text{tb}}}|\Psi_f\rangle, \tag{104}$$

for $x \neq y$ and

$$C_{\uparrow,\Psi}(y,y,t) = \sum_{\{k_z\}\in\{0,1\}} \mathcal{E}_{\Psi_s}(y,\{k_z\}) \langle\Psi_f|e^{itH_{\text{tb}}}\prod_{z=y}^{L} \mathcal{P}_z^{(k_z)}e^{-itH_{\text{tb}}}|\Psi_f\rangle. \tag{105}$$

To write these expressions we used

$$\mathcal{P}_x^{(k)}\mathcal{P}_x^{(p)} = \mathcal{P}_x^{(k)}\delta_{p,k}, \tag{106}$$

and defined

$$\mathcal{C}_{\Psi_s}(x,y,\{k_z\}) \equiv \begin{cases} \langle\Psi_{s,k_{x+1},\dots,k_L}|\mathcal{X}_x^{-1}\sigma_{+,x}\mathcal{X}_{x-1}^{-k_{x-1}}\cdots\mathcal{X}_{y+1}^{-k_{y+1}} \\ \qquad\cdot\sigma_{-,y}\mathcal{X}_y\mathcal{X}_{y+1}^{k_{y+1}}\cdots\mathcal{X}_{x-1}^{k_{x-1}}|\Psi_{s,k_{x+1},\dots,k_L}\rangle \quad x>y \\ \\ \langle\Psi_{s,k_{y+1},\dots,k_L}|\mathcal{X}_{y-1}^{-k_{y-1}}\cdots\mathcal{X}_{x+1}^{-k_{x+1}}\mathcal{X}_x^{-1}\sigma_{+,x} \\ \qquad\cdot\mathcal{X}_{x+1}^{k_{x+1}}\cdots\mathcal{X}_{y-1}^{k_{y-1}}\sigma_{-,y}\mathcal{X}_y|\Psi_{s,k_{y+1},\dots,k_L}\rangle \quad x<y \end{cases}, \tag{107}$$

$$\mathcal{D}_{\Psi_s}(x,y,\{k_z\}) \equiv \begin{cases} \langle\Psi_{s,k_{x+1},\dots,k_L}|\mathcal{X}_{x-1}^{-k_{x-1}}\cdots\mathcal{X}_{y+1}^{-k_{y+1}}\mathcal{X}_y^{-1}\sigma_{-,y} \\ \qquad\cdot\mathcal{X}_{y+1}^{k_{y+1}}\cdots\mathcal{X}_{x-1}^{k_{x-1}}\sigma_{+,x}\mathcal{X}_x|\Psi_{s,k_{x+1},\dots,k_L}\rangle \quad x>y \\ \\ \langle\Psi_{s,k_{y+1},\dots,k_L}|\mathcal{X}_y^{-1}\sigma_{-,y}\mathcal{X}_{y-1}^{-k_{y-1}}\cdots\mathcal{X}_{x+1}^{-k_{x+1}} \\ \qquad\cdot\sigma_{+,x}\mathcal{X}_x\mathcal{X}_{x+1}^{k_{x+1}}\cdots\mathcal{X}_{y-1}^{k_{y-1}}|\Psi_{s,k_{y+1},\dots,k_L}\rangle \quad x<y \end{cases}, \tag{108}$$

$$\mathcal{E}_{\Psi_s}(y,\{k_z\}) \equiv \langle\Psi_{s,k_y,\dots,k_L}|\sigma_{+,y}\sigma_{-,y}|\Psi_{s,k_y,\dots,k_L}\rangle, \tag{109}$$

with

$$|\Psi_{s,k_a,\dots,k_b}\rangle = \mathcal{X}_a^{k_a}\cdots\mathcal{X}_b^{k_b}|\Psi_s\rangle. \tag{110}$$

As shown in Appendix B, the above expressions can be written in the following drastically simpler form

$$\mathcal{C}_{\Psi_s}(x,y,\{k_z\}) = \langle\Psi_{s,K_1,K_2}|I_{\min(x,y)-1}\otimes P^{(1)}_{|y-x|+1,K_1}\otimes I_{L-\max(x,y)}|\Psi_{s,K_1,K_2}\rangle, \tag{111}$$

$$\mathcal{D}_{\Psi_s}(x,y,\{k_z\}) = \langle\Psi_{s,K_1,K_2}|I_{\min(x,y)-1}\otimes P^{(2)}_{|y-x|+1,K_1}\otimes I_{L-\max(x,y)}|\Psi_{s,K_1,K_2}\rangle, \tag{112}$$

$$\mathcal{E}_{\Psi_s}(y,\{k_z\}) = \langle\Psi_{s,0,K_2}|\mathcal{X}_y^{-k_y+1}\sigma_{+,y}\sigma_{-,y}\mathcal{X}_y^{k_y-1}|\Psi_{s,0,K_2}\rangle, \tag{113}$$

where we introduced the rank-1 projectors

$$P^{(1)}_{a,b} = |\underbrace{-\dots-}_{a-b-1}\underbrace{+\dots+}_{b+1}\rangle\langle\underbrace{-\dots-}_{a-b-1}\underbrace{+\dots+}_{b+1}|, \qquad\qquad a\geq b+1, \tag{114}$$

$$P^{(2)}_{a,b} = |\underbrace{+\dots+}_{a-b-1}\underbrace{-\dots-}_{b+1}\rangle\langle\underbrace{+\dots+}_{a-b-1}\underbrace{-\dots-}_{b+1}|, \qquad\qquad a\geq b+1, \tag{115}$$

the state

$$|\Psi_{s,K_1,K_2}\rangle = \mathcal{X}_{\max(x,y)}^{K_1+1} \mathcal{X}_L^{K_2} |\Psi_s\rangle, \tag{116}$$

and we set

$$K_1 \equiv \sum_{z=\min(x,y)+1}^{\max(x,y)-1} k_z, \qquad K_2 \equiv \sum_{z=\max(x,y)+1}^{L} k_z. \tag{117}$$

Using that $k_z = 0, 1$ we find the following ranges for $K_1$ and $K_2$

$$K_1 \in \{0, \ldots, |y-x|-1\}, \qquad K_2 \in \{0, \ldots, L-\max(x,y)\}. \tag{118}$$

The expressions (104, 111, 112) and (105, 113) provide a substantial simplification in the calculation of correlation functions: since the correlators (111)–(113) are constant with respect to time, they can be evaluated once and for all and regarded as fixed coefficients. Given a set of coefficients $\{C_{\Psi_s}(x,y,\{k_z\}), D_{\Psi_s}(x,y,\{k_z\}), \mathcal{E}_{\Psi_s}(y,\{k_z\})\}$ one can then compute the time-evolution of $C_{\uparrow,\Psi}(x,y,t)$ by evaluating a number of correlators in the tight binding model evolving from a Fock state $|\Psi_f\rangle$. Since the latter states are Gaussian, this task can be performed efficiently by writing the free fermion correlators in terms of determinants. Although the sums in (104) and (105) generically involve a very large number of terms (of the order of $2^L$), for specific states (94) and positions $x$ and $y$ the sums can be exactly performed leading to explicit analytical expressions. For instance, let us consider two specific "generalised nested Néel" states of Ref. [82]:

$$|N_{22}\rangle = \prod_{x=1}^{L/4} c_{4x-2,\uparrow}^\dagger c_{4x,\downarrow}^\dagger |0\rangle, \qquad |N_{22}'\rangle = \prod_{x=1}^{L/4} c_{4x-2,\downarrow}^\dagger c_{4x,\uparrow}^\dagger |0\rangle, \tag{119a}$$

$$|N_{22}''\rangle = \prod_{x=0}^{L/4-1} c_{4x+1,\downarrow}^\dagger c_{4x+3,\uparrow}^\dagger |0\rangle, \qquad |N_{22}'''\rangle = \prod_{x=0}^{L/4-1} c_{4x+1,\uparrow}^\dagger c_{4x+3,\downarrow}^\dagger |0\rangle, \tag{119b}$$

and

$$|N_{23}\rangle = \prod_{x=1}^{L/6} c_{6x-4,\uparrow}^\dagger c_{6x-2,\downarrow}^\dagger c_{6x,\downarrow}^\dagger |0\rangle, \qquad |N_{23}'\rangle = \prod_{x=1}^{L/6} c_{6x-4,\downarrow}^\dagger c_{6x-2,\uparrow}^\dagger c_{6x,\uparrow}^\dagger |0\rangle, \tag{120a}$$

$$|N_{23}''\rangle = \prod_{x=0}^{L/6-1} c_{6x+1,\downarrow}^\dagger c_{6x+3,\downarrow}^\dagger c_{6x+5,\uparrow}^\dagger |0\rangle, \qquad |N_{23}'''\rangle = \prod_{x=0}^{L/6-1} c_{6x+1,\uparrow}^\dagger c_{6x+3,\uparrow}^\dagger c_{6x+5,\downarrow}^\dagger |0\rangle. \tag{120b}$$

Note that in the case of (119) we took the chain length to be multiple of 4 while in the case of (120) we took it to be a multiple of 6. We also remark that the dynamics from the states (119) can also be studied with the approach of Ref. [83] but the ones from (120) can not.

### 4.2.1 States (119)

For the initial states (119) we find

$$|N_{22f}\rangle = |N_{22f}'\rangle = |N_f\rangle \equiv \prod_{x=0}^{L/2-1} f_{2x+1}^\dagger |\bigcirc\rangle,$$

$$|N_{22f}''\rangle = |N_{22f}'''\rangle = |\bar{N}_f\rangle \equiv \prod_{x=0}^{L/2-1} f_{2x}^\dagger |\bigcirc\rangle, \tag{121}$$

$$|N_{22s}\rangle = |N_{22s}'''\rangle = |S\rangle \equiv \prod_{y=1}^{L/4} \sigma_{+,2y-1} |-\rangle,$$

$$|N_{22s}\rangle = |N'''_{22s}\rangle = |\bar{S}\rangle \equiv \prod_{y=1}^{L/4} \sigma_{+,2y} |-\rangle \,. \tag{122}$$

From the expressions (121, 122) and the explicit form (111)–(113) of the coefficients one can deduce a number of general constraints. First we note that $\mathcal{C}(x, y, \{k_z\})$ and $\mathcal{D}(x, y, \{k_z\})$ respectively vanish for $K_1 > 1$ and $|x - y| - K_1 < 2$ because the states (122) do not feature a block of up spins of size larger than one. The second observation is that all coefficients vanish if $|x - y| > L/2$. This is because the projectors (114) and (115) can give a non-zero result only if the blocks of down spins in (114)–(115) are contracted with the block of down spins in the states (122). This can happen only if the size of the block of down spins in the states is larger then or equal to the support of the projectors. Noting that the former is $L/2 + 1$ and the latter $|x - y| + 1$ gives the above inequality. Note that this constraint is in agreement with physical intuition: the correlations

$$C_{\uparrow, N_{22}}(x, y, t), \quad C_{\uparrow, N'_{22}}(x, y, t), \quad C_{\uparrow, N''_{22}}(x, y, t), \quad C_{\uparrow, N'''_{22}}(x, y, t), \tag{123}$$

can be non zero only if there are no particles between $x$ and $y$. The best case is then when all other particles are squeezed at the two edges of the system leaving a free region of size $L/2$.

In fact, from (111)–(113) one can easily evaluate the coefficients explicitly. Here, however, we do not present the explicit expressions but rather show two simple examples where the correlations take a particularly simple form.

*Number operators.* The case $x = y$ is perhaps the simplest limiting case. Indeed, recalling the form (122) of the states, we immediately find

$$\begin{aligned} \mathcal{E}_S(y, \{k_z\}) = {} & \delta_{k_y,1} \text{mod}(K_2, 2) \theta(L/2 \geq K_2 + 1) \\ & + \delta_{k_y,0} \text{mod}(K_2 + y - 1, 2) \theta(L/2 \geq K_2 + y), \end{aligned} \tag{124}$$

$$\begin{aligned} \mathcal{E}_{\bar{S}}(y, \{k_z\}) = {} & \delta_{k_y,1} \text{mod}(K_2 + 1, 2) \theta(L/2 > K_2 + 1) \\ & + \delta_{k_y,0} \text{mod}(K_2 + y, 2) \theta(L/2 > K_2 + y). \end{aligned} \tag{125}$$

where we used $0 \leq K_2 \leq L - y$ (cf. (118)). Using now $y > L/2$ we find

$$\mathcal{E}_S(y, \{k_z\}) = \delta_{k_y,1} \text{mod}(K_2, 2), \qquad \mathcal{E}_{\bar{S}}(y, \{k_z\}) = \delta_{k_y,1} \text{mod}(K_2 + 1, 2). \tag{126}$$

Plugging back into (105) and using

$$\sum_{\{k_z\}\in\{0,1\}} \text{mod}(k_{y+1} + \cdots + k_L + a, 2) = \frac{1}{2} \sum_{\eta=\pm} \sum_{\{k_z\}\in\{0,1\}} \eta^{k_{y+1}+\ldots+k_L+a+1}, \tag{127}$$

we find

$$C_{\uparrow, N_{22}}(y, y, t) = \frac{1}{2} \langle N_f | e^{itH_{\text{tb}}} d_y \left[ 1 - \prod_{z=y+1}^{L} (1 - 2d_z) \right] e^{-itH_{\text{tb}}} |N_f\rangle, \tag{128}$$

$$C_{\uparrow, N'_{22}}(y, y, t) = \frac{1}{2} \langle N_f | e^{itH_{\text{tb}}} d_y \left[ 1 + \prod_{z=y+1}^{L} (1 - 2d_z) \right] e^{-itH_{\text{tb}}} |N_f\rangle, \tag{129}$$

$$C_{\uparrow, N''_{22}}(y, y, t) = \frac{1}{2} \langle \bar{N}_f | e^{itH_{\text{tb}}} d_y \left[ 1 + \prod_{z=y+1}^{L} (1 - 2d_z) \right] e^{-itH_{\text{tb}}} |\bar{N}_f\rangle$$

$$= \frac{1}{2} \langle N_f | e^{itH_{tb}} d_{L+1-y} \left[ 1 + \prod_{z=1}^{L-y} (1-2d_z) \right] e^{-itH_{tb}} | N_f \rangle \,, \tag{130}$$

$$C_{\uparrow, N_{22}'''}(y,y,t) = \frac{1}{2} \langle \bar{N}_f | e^{itH_{tb}} d_y \left[ 1 - \prod_{z=y+1}^{L} (1-2d_z) \right] e^{-itH_{tb}} | \bar{N}_f \rangle$$

$$= \frac{1}{2} \langle N_f | e^{itH_{tb}} d_{L+1-y} \left[ 1 - \prod_{z=1}^{L-y} (1-2d_z) \right] e^{-itH_{tb}} | N_f \rangle \,, \tag{131}$$

where in the second lines of (130) and (131) we performed a reflection in the tight-binding model. Using (81), (84) and (86) we then have

$$C_{\uparrow/\downarrow, N_{22}}(y,y,t) = \frac{1}{2} \begin{cases} \langle N_f | e^{itH_{tb}} d_y \left[ 1 \mp \prod_{z=y+1}^{L} (1-2d_z) \right] e^{-itH_{tb}} | N_f \rangle \,, & y > L/2 \\[2em] \langle N_f | e^{itH_{tb}} d_y \left[ 1 \pm \prod_{z=y+1}^{L} (1-2d_z) \right] e^{-itH_{tb}} | N_f \rangle \,, & y \leq L/2 \end{cases} \tag{132}$$

Finally, noting that

$$|N_f\rangle = e^{i\pi D_{\{1,\dots,L\}}} |N_f\rangle = e^{i\pi L} |N_f\rangle \,, \qquad [H_{tb}, D_{\{1,\dots,L\}}] = 0 \,, \tag{133}$$

where $D_A$ is defined in (68), we can rewrite the expressions in the following compact form

$$C_{\uparrow/\downarrow, N_{22}}(y,y,t) = \frac{1}{2} \langle N_f | e^{itH_{tb}} d_y (1 \pm e^{i\pi D_{\{1,\dots,y-1\}}}) e^{-itH_{tb}} | N_f \rangle$$

$$= \frac{1}{2} \langle N_f | e^{itH_{tb}} d_y e^{-itH_{tb}} | N_f \rangle$$

$$\mp \frac{1}{4} \langle N_f | e^{itH_{tb}} (e^{i\pi D_{\{1,\dots,y\}}} - e^{i\pi D_{\{1,\dots,y-1\}}}) e^{-itH_{tb}} | N_f \rangle \,. \tag{134}$$

Using (67) and (69) we then have

$$C_{\uparrow/\downarrow, N_{22}}(y,y,t) = \frac{1}{2} \langle N_f | d_y(t) | N_f \rangle \mp \frac{1}{4} [\chi_N(t, \{1,\dots,y\}, \pi) - \chi_N(t, \{1,\dots,y-1\}, \pi)]$$

$$= \frac{1}{2} \langle N_f | d_y(t) | N_f \rangle \mp \frac{1}{4} \left[ \det(\mathbb{I} - 2\mathbb{C}_{\{1,\dots,y\}}^{N_f}(t)) - \det(\mathbb{I} - 2\mathbb{C}_{\{1,\dots,y-1\}}^{N_f}(t) \right], \tag{135}$$

where the correlation matrix is reported explicitly in (70). Since the characteristic function $\chi_N(t, A, \lambda)$ decays very rapidly to zero after a quench from a state with Néel ordering in the particles (cf. Fig 2), we expect $C_{\uparrow/\downarrow, N_{22}}(y,y,t)$ to rapidly approach $\langle N_f | d_y(t) | N_f \rangle /2$. This is explicitly demonstrated in Fig. 3.

This fact is remarkable: we found that in the presence of Néel ordering in both particles and spin the spin resolved densities approach very rapidly a value that is entirely determined by the free fermion result $\langle N_f | d_y(t) | N_f \rangle$. More precisely we have

$$\langle N_{22} | e^{itH_\infty} n_{x,\uparrow/\downarrow} e^{-itH_\infty} | N_{22} \rangle \mapsto \frac{1}{L} \langle N_{22s} | \sum_{x=1}^{L} \sigma_{+/-,x} | N_{22s} \rangle \langle N_f | d_y(t) | N_f \rangle \,. \tag{136}$$

This means that for these observables the time evolution undergoes two different phases. First they equilibrate "locally" to the time-dependent free fermion result and then they reach stationarity following the free fermionic dynamics. In the next subsection we will see that the same

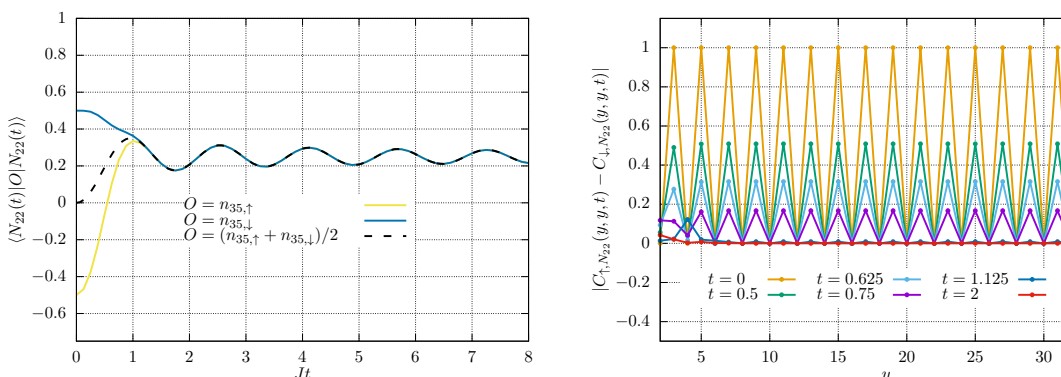

Figure 3: Dynamics of the number operators after a quench from the generalised nested Néel state (119) in the thermodynamic limit. The left panel shows the time evolution of the densities at $x = 35$, comparing $C_{\uparrow,N_{22}}(y,y,t)$, $C_{\downarrow,N_{22}}(y,y,t)$, and $(C_{\uparrow,N_{22}}(y,y,t) + C_{\downarrow,N_{22}}(y,y,t))/2 = \langle N_f | d_y(t) | N_f \rangle /2$. The right panel shows fixed-time slices of $|C_{\uparrow,N_{22}}(y,y,t) - C_{\downarrow,N_{22}}(y,y,t)|$.

kind of "local equilibration" occurs also for generalised Néel ordering in the spin. Namely, this effect takes place every time that the spin configuration $\{\sigma_j\}$ (cf. (48)) has a periodic pattern.

*Boundary Correlations.* Another simple limiting case for the states (122) is $y = L$. Indeed, in this case we have that the range of values that $K_2$ can take is shrunk to $K_2 = 0$ (cf. (118)). Moreover, we immediately see that

$$\mathcal{C}_{\Psi_s}(x, L, \{k_z\}) = \delta_{K_1,0}\delta_{(s)_1,+}\theta(x > L/2), \qquad \mathcal{D}_{\Psi_s}(x, L, \{k_z\}) = 0, \tag{137}$$

where $(s)_1$ is the first spin of $|\Psi_s\rangle$ (one of the two states (122)). Plugging into (104) this leads to

$$C_{\uparrow,\Psi}(x, y, t) = \delta_{(s)_1,+}\theta(L/2 > |x-y|)\langle \Psi_f | e^{itH_{tb}} f_x^\dagger \prod_{z=x+1}^{L-1}(1-d_z)f_L e^{-itH_{tb}} | \Psi_f \rangle. \tag{138}$$

Using again (81), (84) and (86) we finally obtain

$$C_{\uparrow,N_{22}}(x, 1, t) = \theta(x < L/2 + 1)\langle N_f | e^{itH_{tb}} f_x^\dagger \prod_{z=2}^{x-1}(1-d_z)f_1 e^{-itH_{tb}} | N_f \rangle, \tag{139}$$

$$C_{\downarrow,N_{22}}(x, L, t) = \theta(x > L/2)\langle N_f | e^{itH_{tb}} f_x^\dagger \prod_{z=x+1}^{L-1}(1-d_z)f_L e^{-itH_{tb}} | N_f \rangle, \tag{140}$$

$$C_{\uparrow,N_{22}}(x, L, t) = C_{\downarrow,N_{22}}(x, 1, t) = 0. \tag{141}$$

These expressions are again written in terms of determinants involving the correlation matrix (70). In particular, we have

$$\begin{aligned} C_{\uparrow,N_{22}}(x, 1, t) &= \theta(x < L/2 + 1)\det(\mathbb{C}^{N_f'}_{\{1,\dots,x-1\}}(t)), \\ C_{\downarrow,N_{22}}(x, L, t) &= \theta(x > L/2)\det(\mathbb{C}^{N_f''}_{\{x,\dots,L-1\}}(t)), \end{aligned} \tag{142}$$

where we defined

$$[\mathbb{C}^{N_f'}_A]_{x,y} = [\mathbb{C}^{N_f}_A]_{x+1,y} - \delta_{x+1,y}, \qquad [\mathbb{C}^{N_f''}_A]_{x,y} = [\mathbb{C}^{N_f}_A]_{x,y+1} - \delta_{x,y+1}. \tag{143}$$

A representative example of the dynamics of $C_{\uparrow,N}(x, 1, t)$ in the thermodynamic limit is reported in Fig. 4

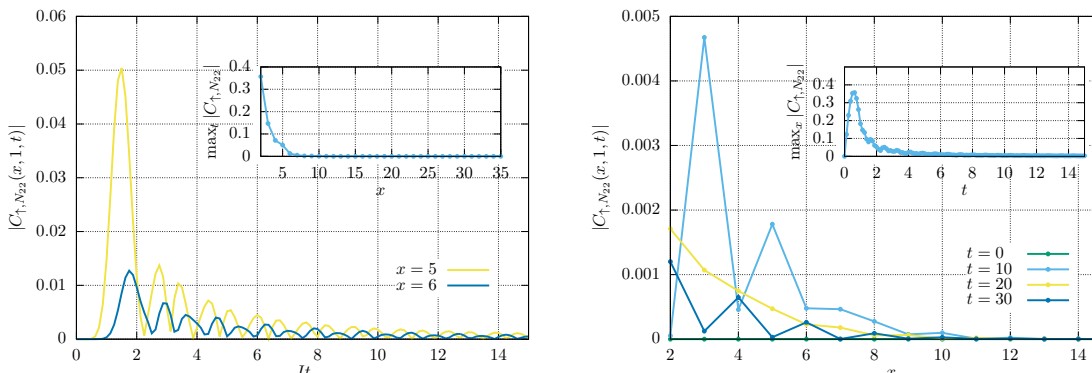

Figure 4: Dynamics of the bulk boundary correlation $C_{\uparrow,N_{22}}(x,1,t)$ after a quench from the nested Néel state (119) in the thermodynamic limit. The left panel reports the time evolution of $|C_{\uparrow,N_{22}}(x,1,t)|$ at two specific positions while the right panel shows four fixed-time cuts of $|C_{\uparrow,N_{22}}(x,1,t)|$ (the insets respectively show the decay in $x$ of $\max_t |C_{\uparrow,N_{22}}(x,1,t)|$ and in $t$ of $\max_x |C_{\uparrow,N_{22}}(x,1,t)|$).

### 4.2.2 States (120)

For the states (120) we find

$$|N_{23f}\rangle = |N'_{23f}\rangle = |N_f\rangle \equiv \prod_{x=0}^{L/2-1} f^\dagger_{2x+1} |\bigcirc\rangle \,,$$

$$|N''_{23f}\rangle = |N'''_{23f}\rangle = |\bar{N}_f\rangle \equiv \prod_{x=0}^{L/2-1} f^\dagger_{2x} |\bigcirc\rangle \,, \tag{144}$$

$$|N_{23s}\rangle = |S_1\rangle \equiv \prod_{y=1}^{L/6} \sigma_{+,3y} |-\rangle \,,$$

$$|N'_{23s}\rangle = |S_2\rangle \equiv \prod_{y=1}^{L/6} \sigma_{+,3y-2}\sigma_{+,3y-1} |-\rangle \,, \tag{145}$$

$$|N''_{23s}\rangle = |\bar{S}_1\rangle \equiv \prod_{y=1}^{L/6} \sigma_{+,3y-2} |-\rangle \,,$$

$$|N'''_{23s}\rangle = |\bar{S}_2\rangle \equiv \prod_{y=0}^{L/6} \sigma_{+,3y-1}\sigma_{+,3y} |-\rangle \,. \tag{146}$$

Once again combining these expressions and the explicit forms (111)–(113) of the coefficients one can deduce a number of general constrains on $K_1$, $K_2$, $x$ and $y$. For instance, we have that all $\mathcal{C}$ coefficients vanish unless $K_1 < 3$ and all $\mathcal{D}$ coefficients vanish unless $|x - y| - K_1 < 4$. Finally, all coefficients are zero if $|x - y| > L/2 + 1$.

Focussing again on simple limiting cases, and using

$$\mathcal{E}_{S_1}(y,\{k_z\}) = \delta_{k_y,1}\delta_{\text{mod}(K_2+1,3)} \,, \tag{147}$$

$$\mathcal{E}_{S_2}(y,\{k_z\}) = \delta_{k_y,1}\delta_{\text{mod}(K_2,3)} + \delta_{k_y,1}\delta_{\text{mod}(K_2+2,3)} \,, \tag{148}$$

$$\mathcal{E}_{\bar{S}_1}(y,\{k_z\}) = \delta_{k_y,1}\delta_{\text{mod}(K_2,3)} \,, \tag{149}$$

$$\mathcal{E}_{\bar{S}_2}(y,\{k_z\}) = \delta_{k_y,1}\delta_{\text{mod}(K_2+1,3)} + \delta_{k_y,1}\delta_{\text{mod}(K_2+2,3)} \,, \tag{150}$$

for $y > L/2$, we find

$$C_{\uparrow,N_{23}}(y,y,t) = \frac{1}{3}\sum_{n=0}^{2} e^{i\frac{2\pi}{3}n} \langle N_f | e^{itH_{tb}} d_y e^{i\frac{2\pi}{3}nD_{\{y+1,\dots,L\}}} e^{-itH_{tb}} | N_f \rangle \,, \tag{151}$$

$$C_{\uparrow,N_{23}'}(y,y,t) = \frac{1}{3}\sum_{n=0}^{2} (1 + e^{i\frac{4\pi}{3}n}) \langle N_f | e^{itH_{tb}} d_y e^{i\frac{2\pi}{3}nD_{\{y+1,\dots,L\}}} e^{-itH_{tb}} | N_f \rangle \,, \tag{152}$$

$$C_{\uparrow,N_{23}''}(y,y,t) = \frac{1}{3}\sum_{n=0}^{2} \langle N_f | e^{itH_{tb}} d_{L-y+1} e^{i\frac{2\pi}{3}nD_{\{1,\dots,L-y\}}} e^{-itH_{tb}} | N_f \rangle \,, \tag{153}$$

$$C_{\uparrow,N_{23}'''}(y,y,t) = \frac{1}{3}\sum_{n=0}^{2} (e^{i\frac{2\pi}{3}n} + e^{i\frac{4\pi}{3}n}) \langle N_f | e^{itH_{tb}} d_{L+1-y} e^{i\frac{2\pi}{3}nD_{\{1,\dots,L-y\}}} e^{-itH_{tb}} | N_f \rangle \,. \tag{154}$$

Employing now (81), (84) and (86) this leads to

$$C_{\uparrow,N_{23}}(y,y,t) = \frac{1}{3}\sum_{n=0}^{2} \langle N_f | e^{itH_{tb}} d_y e^{i\frac{2\pi}{3}nD_{\{1,\dots,y-1\}}} e^{-itH_{tb}} | N_f \rangle \,,$$

$$C_{\downarrow,N_{23}}(y,y,t) = \frac{1}{3}\sum_{n=0}^{2} (e^{i\frac{2\pi}{3}n} + e^{i\frac{4\pi}{3}n}) \langle N_f | e^{itH_{tb}} d_y e^{i\frac{2\pi}{3}nD_{\{1,\dots,y-1\}}} e^{-itH_{tb}} | N_f \rangle \,. \tag{155}$$

In terms of determinants we have

$$C_{\uparrow,N_{23}}(y,y,t) = \frac{1}{3} \langle N_f | e^{itH_{tb}} d_y e^{-itH_{tb}} | N_f \rangle$$
$$+ \frac{1}{3}\sum_{n=1}^{2} \frac{1}{e^{i\frac{2\pi}{3}n}-1} \Big[ \det(\mathbb{I} - (e^{i\frac{2\pi}{3}n}-1)\mathbb{C}_{\{1,\dots,y\}}^{N_f}(t))$$
$$- \det(\mathbb{I} - (e^{i\frac{2\pi}{3}n}-1)\mathbb{C}_{\{1,\dots,y-1\}}^{N_f}(t)) \Big] \,, \tag{156}$$

$$C_{\downarrow,N_{23}}(y,y,t) = \frac{2}{3} \langle N_f | e^{itH_{tb}} d_y e^{-itH_{tb}} | N_f \rangle$$
$$+ \frac{1}{3}\sum_{n=1}^{2} \frac{e^{i\frac{2\pi}{3}n} + e^{i\frac{4\pi}{3}n}}{e^{i\frac{2\pi}{3}n}-1} \Big[ \det(\mathbb{I} - (e^{i\frac{2\pi}{3}n}-1)\mathbb{C}_{\{1,\dots,y\}}^{N_f}(t))$$
$$- \det(\mathbb{I} - (e^{i\frac{2\pi}{3}n}-1)\mathbb{C}_{\{1,\dots,y-1\}}^{N_f}(t)) \Big] \,. \tag{157}$$

As promised, also in this case we see the local equilibration (136) taking place after a short transient.

Moreover, using that the only non-zero coefficients for $y = L$ are

$$\mathcal{C}_{\bar{S}_1}(x,L,\{k_z\}) = \delta_{K_1,0}\theta(x > L/2), \tag{158}$$
$$\mathcal{C}_{S_2}(x,L,\{k_z\}) = (\delta_{K_1,0} + \delta_{K_1,1})\theta(x > L/2 - 1), \tag{159}$$

we find

$$C_{\uparrow,N_{23}}(x,1,t) = \theta(x < L/2 + 1)\det(\mathbb{C}_{\{1,\dots,x-1\}}^{N_f'}(t)),$$

$$C_{\downarrow,N_{23}}(x,L,t) = \theta(x > L/2 - 1)\sum_{y=x+1}^{L-1} \det(\mathbb{C}_{\{x,\dots,L-1\},y}^{N_f''}(t)), \tag{160}$$

where we defined

$$[\mathbb{C}_{A,y_0}^{N_f''}]_{x,y} = \begin{cases} [\mathbb{C}_A^{N_f}]_{x,y+1} - \delta_{x,y+1} & x,y > y_0 \\ [\mathbb{C}_A^{N_f}]_{x,y} - \delta_{x,y} & x > y_0, \quad y \le y_0 \\ [\mathbb{C}_A^{N_f}]_{x-1,y+1} - \delta_{x-1,y+1} & y > y_0, \quad x \le y_0 \\ [\mathbb{C}_A^{N_f}]_{x-1,y} - \delta_{x-1,y} & x,y \le y_0 \end{cases} . \tag{161}$$

This treatment is easily generalised to all states $|N_{pq}\rangle$ with $p,q \ge 2$.

## 5 Conclusions

In this paper we studied the real-time dynamics of the Hubbard model with open boundary conditions in the limit of infinite repulsion. In this limit, as shown in Ref. [86], the Hamiltonian of the model can be unitarily mapped into the tight-binding model at the price of complicating the form of the observables. Here we showed that, in spite of this complication, one can efficiently compute the evolution of the expectation values of certain observables after a quench from initial states in product form. In particular, we pointed out that the expectation value of any function of the total density is exactly equal to the analogous quantity in the tight-binding model. Moreover, we proved that the two point functions of the Hubbard fermions can be expressed as linear combinations of determinants multiplied by simple time-independent coefficients. Specifically, we obtained particularly simple expressions for the expectation value of the densities of particles of each separate species and for the correlation between one point at the boundary and one in the bulk evolving from the generalised nested Néel states of Ref. [82]. Our results on the evolution of total densities are directly extended for initial states not in product form, while generic two point functions become more complicated when foregoing the product structure.

Our results are also the starting point for a number of interesting further developments. First we can use our results to study inhomogeneous initial states such as those corresponding to bipartitioning protocols with global spin imbalance. This would allow to obtain correlation functions which are not accessible in GHD (except for zero entropy states [104]) potentially accessing the diffusive scale. Indeed, the strong coupling Hubbard can be thought of as a quantum version of the classical cellular automaton studied in Refs. [105, 106], which could access the diffusion constant exactly. Another more difficult development concerns the use of our results as a starting point to set up an equations-of-motion scheme — similar to the one used to study prethermalization weakly interacting systems [72, 107–110] — to study the Hubbard model with large but finite interaction. A final extension of our calculations, motivated by recent cold-atom experiments [111] and by the results of Ref. [83], is to apply the generalised Kumar's mapping of Ref. [85] to the quench dynamics of a multispecies Hubbard model with SU(N) symmetry.

## 6 Acknowledgements

We thank Fabian Essler and Lev Vidmar for helpful discussions. BB was supported by the Royal Society through the University Research Fellowship No. 201102. BB thanks the University of Ljubljana for hospitality during the final stages of this project.

## A Kumar mapping on the Hubbard Hamiltonian with finite interaction

To apply the Kumar mapping (9) to the Hamiltonian (1) we break sum across odd and even sites and then write everything in terms of the Majorana fermions

$$H = -J \sum_{\alpha=\uparrow,\downarrow} \left[ \sum_{x=1}^{L/2} \left( c^\dagger_{2x-1,\alpha} c_{2x,\alpha} + c^\dagger_{2x,\alpha} c_{2x-1,\alpha} \right) + \sum_{x=1}^{(L-1)/2} \left( c^\dagger_{2x,\alpha} c_{2x+1,\alpha} + c^\dagger_{2x+1,\alpha} c_{2x,\alpha} \right) \right]$$
$$+ U \sum_{x=1}^{L} \left( n_{x,\uparrow} - \frac{1}{2} \right) \left( n_{x,\downarrow} - \frac{1}{2} \right)$$

$$
\begin{aligned}
&= -J\Bigg[\sum_{x=1}^{L/2}\Big(c^{\dagger}_{2x-1,\uparrow}c_{2x,\uparrow}+c^{\dagger}_{2x,\uparrow}c_{2x-1,\uparrow}\Big)+\sum_{x=1}^{(L-1)/2}\Big(c^{\dagger}_{2x,\uparrow}c_{2x+1,\uparrow}+c^{\dagger}_{2x+1,\uparrow}c_{2x,\uparrow}\Big)\\
&\quad+\sum_{x=1}^{L/2}\Big(c^{\dagger}_{2x-1,\downarrow}c_{2x,\downarrow}+c^{\dagger}_{2x,\downarrow}c_{2x-1,\downarrow}\Big)+\sum_{x=1}^{(L-1)/2}\Big(c^{\dagger}_{2x,\downarrow}c_{2x+1,\downarrow}+c^{\dagger}_{2x+1,\downarrow}c_{2x,\downarrow}\Big)\Bigg]\\
&\quad+U\sum_{x=1}^{L}\Big(n_{x,\uparrow}-\tfrac{1}{2}\Big)\Big(n_{x,\downarrow}-\tfrac{1}{2}\Big)\\
&=-J\sum_{x=1}^{L/2}\big(a^{x}_{2x-1}\sigma^{+}_{2x-1}(-\mathrm{i}a^{y}_{2x}\sigma^{-}_{2x})+\mathrm{i}a^{y}_{2x}\sigma^{+}_{2x}a^{x}_{2x-1}\sigma^{-}_{2x-1}\big)\\
&\quad-J\sum_{x=1}^{(L-1)/2}\big(\mathrm{i}a^{y}_{2x}\sigma^{+}_{2x}a^{x}_{2x+1}\sigma^{-}_{2x+1}+a^{x}_{2x+1}\sigma^{+}_{2x+1}(-\mathrm{i}a^{y}_{2x}\sigma^{-}_{2x})\big)\\
&\quad-\frac{J}{4}\sum_{x=1}^{L/2}\big((\mathrm{i}a^{y}_{2x-1}-a^{x}_{2x-1}\sigma^{z}_{2x-1})(a^{x}_{2x}+\mathrm{i}a^{y}_{2x}\sigma^{z}_{2x})+(a^{x}_{2x}-\mathrm{i}a^{y}_{2x}\sigma^{z}_{2x})(-\mathrm{i}a^{y}_{2x-1}-a^{x}_{2x-1}\sigma^{z}_{2x-1})\big)\\
&\quad-\frac{J}{4}\sum_{x=1}^{(L-1)/2}\big((a^{x}_{2x}-\mathrm{i}a^{y}_{2x}\sigma^{z}_{2x})(-\mathrm{i}a^{y}_{2x+1}-a^{x}_{2x+1}\sigma^{z}_{2x+1})+(\mathrm{i}a^{y}_{2x+1}-a^{x}_{2x+1}\sigma^{z}_{2x+1})(a^{x}_{2x}+\mathrm{i}a^{y}_{2x}\sigma^{z}_{2x})\big)\\
&\quad+\frac{U}{2}\sum_{x=1}^{L}\Big(\tfrac{1}{2}-d_x\Big).
\end{aligned}\tag{162}
$$

Note that in the $U$-dependent term we have applied the identity (16). We then expand and simplify the summands of the first two sums, before combining them with the second two sums. This results in

$$
\begin{aligned}
H=&-J\Bigg[\sum_{x=1}^{L/2}\big(a^{x}_{2x-1}\sigma^{+}_{2x-1}(-\mathrm{i}a^{y}_{2x}\sigma^{-}_{2x})+\mathrm{i}a^{y}_{2x}\sigma^{+}_{2x}a^{x}_{2x-1}\sigma^{-}_{2x-1}\big)\\
&+\sum_{x=1}^{(L-1)/2}\big(\mathrm{i}a^{y}_{2x}\sigma^{+}_{2x}a^{x}_{2x+1}\sigma^{-}_{2x+1}+a^{x}_{2x+1}\sigma^{+}_{2x+1}(-\mathrm{i}a^{y}_{2x}\sigma^{-}_{2x})\big)\\
&+\frac{1}{4}\sum_{x=1}^{L/2}\big((\mathrm{i}a^{y}_{2x-1}-a^{x}_{2x-1}\sigma^{z}_{2x-1})(a^{x}_{2x}+\mathrm{i}a^{y}_{2x}\sigma^{z}_{2x})+(\mathrm{i}a^{y}_{2x-1}+a^{x}_{2x-1}\sigma^{z}_{2x-1})(a^{x}_{2x}-\mathrm{i}a^{y}_{2x}\sigma^{z}_{2x})\big)\\
&+\frac{1}{4}\sum_{x=1}^{(L-1)/2}\big((\mathrm{i}a^{y}_{2x+1}+a^{x}_{2x+1}\sigma^{z}_{2x+1})(a^{x}_{2x}-\mathrm{i}a^{y}_{2x}\sigma^{z}_{2x})+(\mathrm{i}a^{y}_{2x+1}-a^{x}_{2x+1}\sigma^{z}_{2x+1})(a^{x}_{2x}+\mathrm{i}a^{y}_{2x}\sigma^{z}_{2x})\big)\Bigg]\\
&+\frac{U}{2}\sum_{x=1}^{L}\Big(\tfrac{1}{2}-d_x\Big)\\
=&-\frac{J}{2}\Bigg[\sum_{x=1}^{L/2}\big(2a^{x}_{2x-1}\sigma^{+}_{2x-1}(-\mathrm{i}a^{y}_{2x}\sigma^{-}_{2x})+2\mathrm{i}a^{y}_{2x}\sigma^{+}_{2x}a^{x}_{2x-1}\sigma^{-}_{2x-1}+\mathrm{i}a^{y}_{2x-1}a^{x}_{2x}-a^{x}_{2x-1}\sigma^{z}_{2x-1}\mathrm{i}a^{y}_{2x}\sigma^{z}_{2x}\big)\\
&+\sum_{x=1}^{(L-1)/2}\big(2\mathrm{i}a^{y}_{2x}\sigma^{+}_{2x}a^{x}_{2x+1}\sigma^{-}_{2x+1}+2a^{x}_{2x+1}\sigma^{+}_{2x+1}(-\mathrm{i}a^{y}_{2x}\sigma^{-}_{2x})+\mathrm{i}a^{y}_{2x+1}a^{x}_{2x}-a^{x}_{2x+1}\sigma^{z}_{2x+1}\mathrm{i}a^{y}_{2x}\sigma^{z}_{2x}\big)\Bigg]\\
&+\frac{U}{2}\sum_{x=1}^{L}\Big(\tfrac{1}{2}-d_x\Big).
\end{aligned}
$$

The two $J$-dependent sums have the same form, so we combine them introducing a sum over $s=\pm1$. Note that from the boundary conditions $a^{x}_{L+1}=a^{y}_{L+1}=0$, so the term with $x=L/2$, $s=1$ contributes nothing to the sum. The summand has a common factor in the Majorana fermions, so we extract this and simplify the part depending on the spins

$$
H=-\frac{J}{2}\sum_{s=\pm1}\sum_{x=1}^{L/2}\Big[\mathrm{i}a^{y}_{2x}a^{x}_{2x+s}\big(2\sigma^{+}_{2x+s}\sigma^{-}_{2x}+2\sigma^{+}_{2x}\sigma^{-}_{2x+s}+\sigma^{z}_{2x+s}\sigma^{z}_{2x}\big)+\mathrm{i}a^{y}_{2x+s}a^{x}_{2x}\Big]
$$

$$+ \frac{U}{2} \sum_{x=1}^{L} \left( \frac{1}{2} - d_x \right)$$

$$= -\frac{J}{2} \sum_{s=\pm 1} \sum_{x=1}^{L/2} \left[ i a_{2x}^y a_{2x+s}^x (2X_{2x,2x+s} - 1) + i a_{2x+s}^y a_{2x}^x \right] + \frac{U}{2} \sum_{x=1}^{L} \left( \frac{1}{2} - d_x \right),$$

where we introduced the short-hand notation

$$X_{x,x+1} = \frac{1}{2} + \frac{1}{2} \sum_{a=1,2,3} \sigma_{a,x+1} \sigma_{a,x}. \tag{163}$$

Next we express the Hamiltonian in terms of the spinless fermions which are related to the Majorana fermions by (10)

$$H = -\frac{J}{2} \sum_{s=\pm 1} \sum_{x=1}^{L/2} \left[ (f_{2x}^\dagger - f_{2x})(f_{2x+s}^\dagger + f_{2x+s})(\boldsymbol{\sigma}_{2x+s} \cdot \boldsymbol{\sigma}_{2x}) + (f_{2x+s}^\dagger - f_{2x+s})(f_{2x}^\dagger + f_{2x}) \right]$$

$$+ \frac{UL}{4} - \frac{U}{2} \sum_{x=1}^{L} f_x^\dagger f_x$$

$$= -J \sum_{s=\pm 1} \sum_{x=1}^{L/2} \left[ (f_{2x}^\dagger f_{2x+s} + f_{2x+s}^\dagger f_{2x}) X_{2x,2x+s} + (f_{2x}^\dagger f_{2x+s}^\dagger - f_{2x} f_{2x+s})(X_{2x,2x+s} - 1) \right]$$

$$- \frac{U}{2} \sum_{x=1}^{L} \left( f_x^\dagger f_x - \frac{1}{2} \right).$$

Finally, we sum explicitly over $s$ and observe that we can rewrite the sum over all the sites instead of having separate terms for odd and even sites. This leads to (18).

## B   Simplification of the Coefficients

Using the algebraic relations

$$\mathcal{X}_a^{-1} \sigma_{\pm, x \bmod a} \mathcal{X}_a = \sigma_{\pm, x+1 \bmod a}, \quad \mathcal{X}_a^{-1} (O_k \otimes I_{a-k}) \mathcal{X}_a = \mathcal{X}_b^{-1} (O_k \otimes I_{b-k}) \mathcal{X}_b, \tag{164}$$

where $O_k$ is a generic operator of support $k$ and $a, b > k$, together with

$$\mathcal{X}_a^{-1} \mathcal{X}_b = \mathcal{X}_{\max(a,b)}^{-1} (I_{\min(a,b)-1} \otimes \mathcal{X}_{|b-a|+1}^{\mathrm{sgn}(b-a)}) \mathcal{X}_{\max(a,b)}, \tag{165}$$

we can rewrite (107) and (108) as follows

$$C_{\Psi_s}(x, y, \{k_z\}) = \begin{cases} \langle \Psi_{s,K_1,K_2} | \sigma_{+,x-K_1} (I_{y-1} \otimes \mathcal{X}_{x-y+1}^{-1}) \sigma_{-,x} | \Psi_{s,K_1,K_2} \rangle & x > y \\ \\ \langle \Psi_{s,K_1,K_2} | \sigma_{+,y} (I_{x-1} \otimes \mathcal{X}_{y-x+1}) \sigma_{-,y-K_1} | \Psi_{s,K_1,K_2} \rangle & x < y \end{cases}, \tag{166}$$

$$\mathcal{D}_{\Psi_s}(x, y, \{k_z\}) = \begin{cases} \langle \Psi_{s,K_1,K_2} | \sigma_{-,x} (I_{y-1} \otimes \mathcal{X}_{x-y+1}) \sigma_{+,x-K_1} | \Psi_{s,K_1,K_2} \rangle & x > y \\ \\ \langle \Psi_{s,K_1,K_2} | \sigma_{-,y-K_1} (I_{x-1} \otimes \mathcal{X}_{y-x+1}^{-1}) \sigma_{+,y} | \Psi_{s,K_1,K_2} \rangle & x < y \end{cases}, \tag{167}$$

$$\mathcal{E}_{\Psi_s}(y, \{k_z\}) = \langle \Psi_{s,0,K_2} | \mathcal{X}_y^{-k_y+1} \sigma_{+,y} \sigma_{-,y} \mathcal{X}_y^{k_y-1} | \Psi_{s,0,K_2} \rangle, \tag{168}$$

where $|\Psi_{s,K_1,K_2}\rangle$ is defined in Eq. (116) while $K_1$ and $K_2$ are defined in Eq. (118). Now we note that $|\Psi_{s,K_1,K_2}\rangle$ is an eigenvector of $\sigma_{3,x}$ for all $x$. Namely it is written as

$$|\Psi_{s,K_1,K_2}\rangle = |s_1,\ldots,s_L\rangle\,, \tag{169}$$

for some $\{s_j\} = \pm$. This follows from the fact that $|\Psi_s\rangle$ is of the form (169) and $\mathcal{X}_z$ are deterministic in the basis (169).

Now we consider $\mathcal{C}_{S_A}(x,y,\{k_z\})$ and $\mathcal{D}_{S_A}(x,y,\{k_z\})$ separately. In particular, since

$$\langle\Psi_{s,K_1,K_2}|\sigma_{+,\max(y,x)}(I_{\min(y,x)-1}\otimes\mathcal{X}_{|y-x|+1})\sigma_{-,\max(y,x)-K_1}|\Psi_{s,K_1,K_2}\rangle =$$
$$= \prod_{i=1}^{\max(x,y)-K_1-1}\delta_{s_i,-}\prod_{i=\max(x,y)-K_1}^{\max(x,y)}\delta_{s_i,+}\,, \tag{170}$$

we have

$$\mathcal{C}_{\Psi_s}(x,y,\{k_z\}) = \langle\Psi_{s,K_1,K_2}|I_{\min(x,y)-1}\otimes P^{(1)}_{|y-x|+1,K_1}\otimes I_{L-\max(x,y)}|\Psi_{s,K_1,K_2}\rangle\,. \tag{171}$$

where $P^{(1)}_{a,b}$ is defined in (114). Similarly, we have

$$\langle\Psi_{s,K_1,K_2}|\sigma_{-,\max(y,x)-K_1}(I_{\min(y,x)-1}\otimes\mathcal{X}^{-1}_{|y-x|+1})\sigma_{+,\max(y,x)}|\Psi_{s,K_1,K_2}\rangle =$$
$$= \prod_{i=1}^{\max(x,y)-K_1-1}\delta_{s_i,-}\prod_{i=\max(x,y)-K_1}^{\max(x,y)}\delta_{s_i,+}\,, \tag{172}$$

which implies

$$\mathcal{D}_{\Psi_s}(x,y,\{k_z\}) = \langle\Psi_{s,K_1,K_2}|I_{\min(x,y)-1}\otimes P^{(2)}_{|y-x|+1,K_1}\otimes I_{L-\max(x,y)}|\Psi_{s,K_1,K_2}\rangle\,, \tag{173}$$

where $P^{(2)}_{a,b}$ is defined in (115).

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
