# Peer review of "Real-Time Evolution in the Hubbard Model with Infinite Repulsion"

_SciPost Physics, doi:SciPost Phys. 12, 028 (2022)_

## Round 1 · Referee Report · Anonymous (Referee 1) · 2021-11-23

Report

In this work authors study the time evolution in the Hubbard model following the quantum quench. Authors focus on the infinite repulsion limit in which the model can be mapped into a non-interacting theory, through a Kumar's transformation. Whereas the resulting Hamiltonian is simple the relation between the operators of initial and final theories is more intricate and therefore the time evolution of expectation values is certainly worth exploring.

The results presented by the authors concern two classes of operators: (i) arbitrary analytic function of arbitrary number of local number operators and (ii) quadratic monomials of creation, annihilation operators. As initial states they consider kinds of Neel states. The discussion of the results from the physical perspective is minimal.

The results are interesting and the paper is well written. Before suggesting the publication I would like the authors to fix some small issues that I list below.

Requested changes

  • In section 4.1 concerning the expectation values of total number of spinfull fermions the authors consider $C_{\Psi}$, eq. (4.15), which contains operators not of this form. Please comment.

  • Could the the authors be more explicit with explaining the step leading from (4.48) to (4.50). Especially, please expand the phrase between (4.49) and (4.50).

  • Does the $\in$ symbol (4.69) and (4.70) refer to the values of $K_i$? If so, it's better to split the definition of $K_i$ from their properties. Otherwise, please rewrite because the meaning is not clear.

Typos:

  • in the text between (3.17) and (3.18): t should be J.
  • is $N$ missing in $C_{\Psi_N}$ and $D_{\Psi_N}$ of eqs. (4.15)?
  • no $N$ on the l.h.s of eqs. (4.16).
  • constrains -> constraints, below (4.74).
  • "decays very rapidly to zero", below eq. (4.87)
  • double 'of the' in the conclusions.

---

## Round 1 · Referee Report · Anonymous (Referee 2) · 2021-11-25

Strengths

  • exact calculations
  • all steps are well explained

Weaknesses

  • lack of test for the results
  • physical arguments lacking

Report

This is an interesting paper, where all calculations are clearly conducted on a professional level, and they are correct, but it would be nice to see some comparison with another approach (a numerical method for example). The main problem I see is the lack of a more global picture. We understand that solving the dynamics in this model is non-trivial due to the mapping which is non-local, but what do we learn from these calculations? For example, it would be intriguing to understand how to obtain the large time correlations after a quench: is there a GGE only expressed in terms of the free fermion correlations, or it is more complicated than that? And what about the thermodynamic limit of the overlap functions? And anyway, how the dynamic of infinite U-Hubbard then differ from a free fermion dynamics? It would be physically-relevant to compare the plots in the paper with a free fermion calculation to see this. The manuscript would greatly benefit from such discussions.

Requested changes

In order of priority
-- add some numerical check, if possible
-- discuss GGE and steady states after quenches
--compare the result with the one of a free fermion theory

---

## Round 2 · Referee Report · Anonymous (Referee 1) · 2021-12-2

Report

The authors addressed my concerns.

---

## Round 2 · Author Response

We thank both referees for their careful reading of our manuscript, for their relevant comments, and for their overall positive assessment. We have made a number of modifications to the manuscript to accomodate their comments. To help identifying the changes we highlighted them in red in the new version.

Response to Referee 1

We thank the referee for finding our paper interesting and clear and for providing us with constructive feedback. In the following we proceed to address the main criticisms of the referee, i.e.,

1) lack of testing for the results" 2)lack of physical arguments".

in a separate fashion.

Concerning the first point, even though we understand the reasoning the referee, we disagree with their conclusions and accordingly we decided not to add numerical tests. This for two main reasons. First, a philosophical one: our results are exact, there is no approximation that one should check. A numerical check could merely prove that we did not make mistakes in our algebraic manipulations (which the referee confirmed are sound). Second, obtaining our results numerically is very hard and can be done only for short times. Indeed, even the most advanced numerical techniques available to stimulate non-equilibrium dynamics for quantum many-body systems in 1d, i.e. those based on tensor-network techniques, are severely hampered by the growth of entanglement and are then limited to short times. In particular, for the problem at hand we will not be able to reach times larger than $20J$ (possibly less). This is actually one of the main points of interest of our paper and to highlight it we decided to slightly modify the discussion in the introduction.

Concerning the second point we partially agree with the spirit of the comment and we accordingly expanded some of the discussions making more obvious the comparison with the free-fermion results (which, as a matter of fact, was already present). In particular we expanded the discussions after Eqs. (4.12), (4.88), (4.110). Let us point out, however, that this paper is only focussed in the finite-time dynamics, not on the infinite time limit. A thorough description of the stationary state and its properties is presented in our previous paper on the topic, i.e., Ref. [77].

Response to Referee 2

We sincerely thank the referee for their very thorough reading of our manuscript, for their positive assessment, and for their useful comments and suggestions. Below we address them point-by-point

"- In section 4.1 concerning the expectation values of total number of spinfull fermions the authors consider $C_\Psi$, eq. (4.15), which contains operators not of this form. Please comment."

The referee is right. However $C_{\Psi_N}(x,x,t)$, which is the one used in Eq.~(4.14), is indeed of the right form. We added a brief comment in the new version.

" - Could the the authors be more explicit with explaining the step leading from (4.48) to (4.50). Especially, please expand the phrase between (4.49) and (4.50).} "

In that step we used (4.49) and the fact that $[f^{(\dag)}_x,U^{(\dag)}_y]=0$ for $x\neq y$ to replace $U^\dag_x$ and $U^\dag_y$ on the left of $f_x^\dag f_y$ with ${\cal X}^\dag_x$ and $I$ respectively. We also replaced $U_x$ and $U_y$ on the right with $I$ and ${\cal X}_y$. This gives the first line of (4.50). Analogous reasonings give the second line. In the new version we expanded the explanation as requested.

"- Does the $\in$ symbol (4.69) and (4.70) refer to the values of $K_i$? If so, it's better to split the definition of $K_i$ from their properties. Otherwise, please rewrite because the meaning is not clear."

Yes, it does. We agree with the referee: in the new version we performed the requested split.

"Typos: - in the text between (3.17) and (3.18): t should be J. - is $N$ missing in $C_{\Psi_N}$ and $D_{\Psi_N}$ of eqs. (4.15)? - no $N$ on the l.h.s of eqs. (4.16). - constrains $\rightarrow$ constraints, below (4.74). - ``decays very rapidly to zero", below eq. (4.87) - double 'of the' in the conclusions."

We thank the referee for signalling these typos. They are all fixed in the new version.

---

## Editorial Decision

published